# FOSI: Hybrid First and Second Order Optimization

**Hadar Sivan**
Technion
Haifa, Israel
hadarsivan@cs.technion.ac.il

**Moshe Gabel**
York University
Toronto, Canada
mgabel@yorku.ca

**Assaf Schuster**
Technion
Haifa, Israel
assaf@cs.technion.ac.il

## Abstract

Popular machine learning approaches forgo second-order information due to the difficulty of computing curvature in high dimensions. We present FOSI, a novel meta-algorithm that improves the performance of any base first-order optimizer by efficiently incorporating second-order information during the optimization process. In each iteration, FOSI implicitly splits the function into two quadratic functions defined on orthogonal subspaces, then uses a second-order method to minimize the first, and the base optimizer to minimize the other. We formally analyze FOSI's convergence and the conditions under which it improves a base optimizer. Our empirical evaluation demonstrates that FOSI improves the convergence rate and optimization time of first-order methods such as Heavy-Ball and Adam, and outperforms second-order methods (K-FAC and L-BFGS).

## 1 Introduction

Consider the optimization problem $\min_\theta f(\theta)$ for a twice differential function $f : \mathbb{R}^n \to \mathbb{R}$. First-order optimizers such as gradient descent (GD) use only the gradient information to update $\theta$ (Kingma & Ba, 2014; Tieleman et al., 2012; Duchi et al., 2011; Polyak, 1987; Nesterov, 2003). Conversely, second-order optimizers such as Newton's method update $\theta$ using both the gradient and the Hessian information. First-order optimizers are thus more computationally efficient as they only require evaluating and storing the gradient, and since their update step often involves only element-wise operations, but have a lower convergence rate compared to second-order optimizers in many settings (Tan & Lim, 2019). Unfortunately, second-order optimizers cannot be used for large-scale optimization problems such as deep neural networks (DNNs) due to the intractability of evaluating the Hessian when the dimension $n$ is large.

Despite recent work on hybrid optimizers that leverage second-order information without computing the entire Hessian (Henriques et al., 2019; Martens & Grosse, 2015; Gupta et al., 2018; Goldfarb et al., 2020; Sivan et al., 2022), first-order methods remain the preferred choice for two reasons. First, many hybrid methods approximate the Hessian rather than the inverse preconditioner directly, resulting in amplifying approximation error and noise (Li, 2017). Second, no single optimizer is best across all problems: the performance of an optimizer can depend on the specific characteristics of the problem it is being applied to (Nocedal & Wright, 1999; Wilson et al., 2017; Zhou et al., 2020).

**Our Contributions.** We propose FOSI (for **F**irst-**O**rder and **S**econd-order **I**ntegration), an alternative approach. Rather than creating a completely new optimizer, FOSI improves the convergence of any base first-order optimizer by incorporating second-order information. FOSI iteratively splits $\min_\theta f(\theta)$ into pairs of quadratic problems on orthogonal subspaces, then uses Newton's method to optimize one and the base optimizer to optimize the other. Unlike prior approaches, FOSI: (a) estimates the inverse preconditioner directly, reducing errors due to matrix inversion; (b) only estimates the most extreme eigenvalues and vectors, making it more robust to noise; (c) has low and controllable overhead; (d) accepts a base first-order optimizer, making it well suited for a large variety of tasks; and (e) works as "turn key" replacement for the base optimizer without requiring additional tuning. We make the following contributions:

- A detailed description of the FOSI algorithm and a thorough spectral analysis of its preconditioner. We prove FOSI converges under common assumptions, and that it improves the condition number of the problem for a large family of base optimizers.

- An empirical evaluation of FOSI on common DNN training tasks with standard datasets, showing it improves over popular first-order optimizers in terms of convergence and wall time. The best FOSI optimizer achieves the same loss as the best first-order algorithm in 48%–77% of the wall time, depending on the task. We also use quadratic functions to explore different features of FOSI, showing it significantly improves convergence of base optimizers when optimizing ill-conditioned functions with non-diagonally dominant Hessians.
- An open source implementation of FOSI, available at: `https://github.com/hsivan/fosi`.

## 2 BACKGROUND AND NOTATION

Given $\theta_t$, the parameter vector at iteration $t$, second-order methods incorporate both the gradient $g_t = \nabla f(\theta_t)$ and the Hessian $H_t = \nabla^2 f(\theta_t)$ in the update step, while first-order methods use only the gradient. These algorithms typically employ an update step of the form $\theta_{t+1} = \theta_t + d_t$, where $d_t$ is a descent direction determined by the information (first and/or second order) from current and previous iterations. Usually, $d_t$ is of the form $-\eta P_t^{-1} \bar{g}_t$, where $P_t$ is a preconditioner matrix, $\bar{g}_t$ is a linear combination of current and past gradients, and $\eta > 0$ is a learning rate. This results in an effective condition number of the problem given by the condition number of $P_t^{-1} H_t$, which ideally is smaller than that of $H_t$ (Zupanski, 2002). Note that in Newton's method $P_t = H_t$, resulting in the ideal effective condition number of 1.

Since evaluating $H_t$ for large $n$ is intractable, most prior work approximate it. This, however, results in amplification of approximation errors and gradient noise due to the need of computing the inverse $P_t^{-1}$ (Li, 2017); techniques such as damping (Martens & Grosse, 2015) that artificially modify the approximated Hessian further increase the error. As we will later show, FOSI approximates $P_t^{-1}$ directly, which avoids the error amplification induced by matrix inversion.

**The Lanczos algorithm.** To obtain information about the curvature of a function $f$ without computing its entire Hessian, we use the Lanczos algroithm (1950). This is an iterative method that finds the $m$ extreme eigenvalues and eigenvectors of a symmetric matrix $A \in \mathbb{R}^{n \times n}$, where $m$ is usually much smaller than $n$. After running $m$ iterations, its output is a matrix $U \in \mathbb{R}^{n \times m}$ with orthonormal columns and a tridiagonal real symmetric matrix $T \in \mathbb{R}^{m \times m}$ which can then be used to extract the approximate eigenvalues and eigenvectors of $A$.

The Lanczos approximation is more accurate for more extreme eigenvalues, thus to accurately approximate the $k$ largest and $\ell$ smallest eigenvalues, $m$ must be larger than $k + \ell$. Crucially, the Lanczos algorithm does not require storing $A$ explicitly. It only requires an operator that receives a vector $\mathbf{v}$ and computes the matrix-vector product $A\mathbf{v}$. In our case, $A$ is the Hessian $H_t$ of $f(\theta)$ at the point $\theta_t$. We denote by $hvp_t(\mathbf{v}) : \mathbb{R}^n \to \mathbb{R}$ the operator that returns the Hessian vector product $H_t \mathbf{v}$. This operator can be evaluated in linear time (roughly two approximations of $f$'s gradient), using Pearlmutter's algorithm (1994).

**Notations and definitions.** We use diag($\mathbf{v}$) for a diagonal matrix with diagonal $\mathbf{v}$, $\mathbf{0}_m$ or $\mathbf{1}_m$ for a row vector of zeros or ones of size $m$, and $[A, B]$ for concatenating two matrices $n \times m_1$, $n \times m_2$ into a single $n \times (m_1 + m_2)$ matrix. We also define several notations w.r.t a real symmetric matrix $A$ with eigenvalues $\lambda_1 > ... > \lambda_n$ and eigenvectors $\mathbf{v}_1, ..., \mathbf{v}_n$. Let $\widehat{\boldsymbol{\lambda}}$ be the row vector with entries $\lambda_1, ..., \lambda_k$ and $\lambda_{n-\ell+1}, ..., \lambda_n$ (the $k$ largest and $\ell$ smallest eigenvalues), and $\widehat{V} \in \mathbb{R}^{n \times k+\ell}$ the corresponding matrix whose columns are the eigenvectors of the eigenvalues in $\widehat{\boldsymbol{\lambda}}$. Similarly, $\widecheck{\boldsymbol{\lambda}}$ is the row vector $[\lambda_{k+1}, ..., \lambda_{n-\ell}]$ and $\widecheck{V} \in \mathbb{R}^{n \times n-k-\ell}$ is the corresponding matrix of eigenvectors.

## 3 FIRST AND SECOND-ORDER INTEGRATION

FOSI is a hybrid method that combines a first-order base optimizer with Newton's method by utilizing each to operate on a distinct subspace of the problem. The Lanczos algorithm, which provides curvature information of a function, is at the core of FOSI. We first provide an algorithm for approximating extreme eigenvalues and eigenvectors (§3.1). We next present FOSI (§3.2), analyze its preconditioner (§3.3), discuss use of momentum (§3.4), and analyze convergence in stochastic settings such as DNN training (§3.5). We then discuss support for closed-form learning rates (§3.6), reducing spectrum estimation error, and FOSI's overhead (§3.7).

### 3.1 Extreme Spectrum Estimation (ESE)

FOSI uses the Lanczos algorithm to estimate the extreme eigenvalues and vectors of the Hessian $H_t$. Recently, Urschel (2021) presented probabilistic upper and lower bounds on the relative error of this approximation for arbitrary eigenvalues. While the upper bound is dependent on the true eigenvalues of $H_t$, which is unknown, the lower bound is dependent solely on $m$ and $n$. To maintain the lower bound small, it is necessary to set $m$ such that $m = \Theta(\ln n)$ and $m$ must be greater than $k + \ell$. We thus define a heuristic for determining $m$: $m = \max\{4(k + \ell), 2 \ln n\}$.

We now describe the ESE procedure for obtaining the $k$ largest and $\ell$ smallest eigenvalues of $H_t$ and their eigenvectors using Lanczos. ESE takes as input the function $f$ and its parameter value $\theta_t$, and uses them to define a Hessian-vector product operator $hvp_t$. Next, it calls the Lanczos algorithm with a specified number of iterations, $m$, and the $hvp_t$ operator. Our implementation parallelizes $hvp_t$ computations across the batch dimension, since they involve gradient computation, and performs full orthogonalization w.r.t all previous vectors in each iteration to prevent numerical instability (Meurant & Strakoš, 2006). Finally, ESE extracts the desired eigenvalues and eigenvectors from Lanczos's outputs. The steps are summarized as Algorithm 1 in the Supplementary Material (Appendix A.1).

### 3.2 The FOSI Optimizer

FOSI takes as input the base optimizer, the function to be optimized, and an initial point, then performs iterative updates until convergence is reached. In each iteration $t$, FOSI computes the gradient $g_t = \nabla f(\theta_t)$, potentially updates the spectrum estimation, then uses both to update $\theta_t$.

FOSI calls the ESE procedure every $T \geq 1$ iterations to obtain $\widehat{\lambda}$ and $\widehat{V}$, the largest $k$ and smallest $\ell$ eigenvalues of $H_t$ and their eigenvectors, and then computes $\mathbf{u} = 1/|\widehat{\lambda}|$ using element-wise absolute values. To avoid approximation errors, we postpone the first invocation of the ESE procedure by $W$ warmup iterations (we discuss this further in §3.7) During these iterations, the updates are equivalent to those of the base optimizer, as $\mathbf{u}$ and $\widehat{V}$ are initialized as zeros.

Next, FOSI updates $\theta_t$ using the following procedure:

1. Compute the sum of $g_t$'s projections on $\widehat{V}$'s columns, $g_1 = \widehat{V}(\widehat{V}^T g_t)$, and the sum of $g_t$'s projections on $\widetilde{V}$'s columns, $g_2 = g_t - g_1$. Due to the orthogonality of the eigenvectors, $g_1$ and $g_2$ are also orthogonal to each other.
2. Compute the descent direction $d_1 = -\alpha \widehat{V}((\widehat{V}^T g_1) \odot \mathbf{u}^T)$, where $\odot$ stands for the Hadamard product. While the chance of encountering an eigenvalue that is exactly or nearly 0 when using small $k$ and $\ell$ values is very small, it is common to add a small epsilon to $|\widehat{\lambda}|$ to avoid division by such values (Kingma & Ba, 2014) when computing $\mathbf{u}$. Note that an equivalent computation to $d_1$ is $-\alpha \widehat{V} \text{diag}(\mathbf{u}) \widehat{V}^T g_t$, which is an $\alpha$-scaled Newton's method step that is limited to $\widehat{V}$ subspace. The resulting $d_1$ is a linear combination of $\widehat{V}$'s columns.
3. Call the base optimizer to compute a descent direction from $g_2$, denoted by $d_b$.
4. Subtract from $d_b$ its projection on $\widehat{V}$'s columns, $d_2 = d_b - \widehat{V}(\widehat{V}^T d_b)$. The new vector $d_2$ is orthogonal to $\widehat{V}$'s columns, hence also to $d_1$.
5. Update the parameters: $\theta_{t+1} = \theta_t + d_1 + d_2$

Parentheses in the above steps are important as they allow for only matrix-vector products, reducing computational complexity. Appendix A.2 provides the full pseudocode for FOSI.

**Splitting to Two Subspaces.** For clarity, we define $\omega = \theta_t - \theta$, $H_1 = \widehat{V} \text{diag}(\widehat{\lambda}) \widehat{V}^T$, and $H_2 = \widetilde{V} \text{diag}(\widetilde{\lambda}) \widetilde{V}^T$. Then at each iteration $t$, FOSI implicitly uses the quadratic approximation $\tilde{f} = f_t + \omega^T g_t + \frac{1}{2} \omega^T H_t \omega$ of $f$ and performs a step to minimize $\tilde{f}$ as follows. It first divides the vector space that is the eigenvectors of $H_t$ into two orthogonal complement subspaces – one is spanned by $\widehat{V}$'s columns and the other by $\widetilde{V}$. It then implicitly splits $\tilde{f}$ into two quadratic functions $f_1$ and $f_2$ such that $\tilde{f}$ is their sum: $f_1 = \frac{1}{2} f_t + \omega^T g_1 + \frac{1}{2} \omega^T H_1 \omega$ and $f_2 = \frac{1}{2} f_t + \omega^T g_2 + \frac{1}{2} \omega^T H_2 \omega$. Note that $\tilde{f} = f_1 + f_2$, since $g_t = g_1 + g_2$ and $H_t = H_1 + H_2$. Finally, FOSI minimizes $f_1$ and $f_2$ independently, while using a scaled Newton's step to minimize $f_1$ and the base optimizer step to minimize $f_2$.

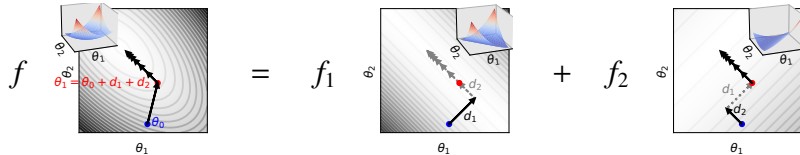

Figure 1: FOSI's update steps (arrows) when minimizing a quadratic function $f(\theta)$. FOSI implicitly separates the space into two orthogonal complement subspaces and then splits the original function $f$ into two functions $f_1$ and $f_2$ over these subspaces, such that $f = f_1 + f_2$. FOSI solves $\min f$ by simultaneously solving $\min f_1$ with Newton's method and $\min f_2$ with the base optimizer. The update step is the sum of $d_1$ and $d_2$, the updates to $f_1$ and $f_2$ respectively.

We observe that $f_1$ has similar slope and curvature as $\tilde{f}$ in the subspace that is spanned by $\widehat{V}$ and zero slope and curvature in its orthogonal complement $\breve{V}$, while $f_2$ has similar slope and curvature as $\tilde{f}$ in $\breve{V}$ and zero slope and curvature in $\widehat{V}$. To minimize $f_1$, FOSI changes $\theta$ in the direction $d_1$ that is a linear combination of $\widehat{V}$'s columns, and to minimize $f_2$, it changes $\theta$ in the direction $d_2$ that is a linear combination of $\breve{V}$'s columns. Hence, we can look at each step of FOSI as two simultaneous and orthogonal optimization steps that do not affect the solution quality of each other, and their sum is a step in the minimization of $\tilde{f}$. Figure 1 illustrates this idea for the quadratic function $f(\theta) = 1.25\theta_1^2 + 1.25\theta_2^2 + 1.5\theta_1\theta_2$.

**Avoiding Matrix Inversion.** We stress that unlike most hybrid methods, FOSI does not require inverting $H_1$. Rather, the inverse preconditioner is obtained directly and exactly using the output of ESE: $H_1^{-1} = \widehat{V} \operatorname{diag}(\mathbf{u})\widehat{V}^T$. This helps avoid error amplification due to matrix inversion (Li, 2017).

### 3.3 Preconditioner Analysis

We next analyze FOSI as a preconditioner. For simplicity, the $t$ subscript is omitted from $g_t$ and $H_t$ when it is clear from the text that the reference is to a specific point in time.

For base optimizers that utilize a diagonal matrix as a preconditioner (e.g., Adam), the result is an efficient computation, as $P^{-1}g$ is equivalent to element-wise multiplication of $P^{-1}$'s diagonal with $g$. When using FOSI with such a base optimizer, the diagonal of the inverse preconditioner, denoted by $\mathbf{q}$, is calculated using $g_2$, instead of $g$. Hence, $d_b = -\eta \operatorname{diag}(\mathbf{q})g_2$, for some learning rate $\eta > 0$.

**Lemma 1.** *Let $f(\theta)$ be a convex twice differential function and let BaseOpt be a first-order optimizer that utilizes a positive definite diagonal preconditioner. Let $H$ be $f$'s Hessian at iteration $t$ of FOSI with BaseOpt, and let $V \operatorname{diag}(\boldsymbol{\lambda})V^T$ be an eigendecomposition of $H$ such that $V = [\widehat{V}, \breve{V}]$ and $\boldsymbol{\lambda} = [\widehat{\boldsymbol{\lambda}}, \breve{\boldsymbol{\lambda}}]$. Then:*

1. *FOSI's inverse preconditioner is $P^{-1} = V \begin{pmatrix} \alpha \operatorname{diag}(\mathbf{u}) & \mathbf{0} \\ \mathbf{0} & \eta M \end{pmatrix} V^T$, where $M$ is the trailing $n-k-\ell$ principal submatrix (lower right corner submatrix) of $V^T \operatorname{diag}(\mathbf{q})V$, and $\operatorname{diag}(\mathbf{q})$ is the inverse preconditioner produced by BaseOpt from $g_2$.*
2. *The preconditioner $P$ is symmetric and positive definite.*
3. *$\alpha$ is an eigenvalue of the effective Hessian $P^{-1}H$, and $\widehat{V}$'s columns are in the eigenspace of $\alpha$.*

The proof can be found in Appendix A.3. It includes expressing $d_1$ and $d_2$ as a product of certain matrices with $g$, substituting these expressions into $\theta_t + d_1 + d_2$, and using linear algebra properties to prove that the resulting preconditioner is a symmetric positive definite matrix. Finally, we assign $P^{-1}$ expression to obtain $P^{-1}H$. Note that symmetric positive definite preconditioner is necessary for ensuring that the search direction always points towards a descent direction (Li, 2017).

As expected, we obtained a separation of the space into two subspaces. For the subspace that is spanned by $\widehat{V}$, for which FOSI uses scaled Newton's method, the condition number is 1. For the complementary subspace, the condition number is determined by BaseOpt's preconditioner. In the general case, it is hard to determine the impact of a diagonal preconditioner on the condition number of the problem, although it is known to be effective for diagonally dominant Hessian (Qu et al., 2020;

Levy & Duchi, 2019). Appendix A.3 includes an analysis of the special case in which $H$ is diagonal. We show that even in this case, which is ideal for a diagonal preconditioner, FOSI provides benefit, since it solves $f_1$ with Newton's method and provides the base optimizer with $f_2$, which is defined on a smaller subspace, hence can be viewed as of smaller dimensionality than $f$.

**Identity Preconditioner.** For base optimizers with identity preconditioner such as GD, we can perform a complete spectral analysis of FOSI's preconditioner and effective Hessian, even for non-diagonal $H$. This allows us to obtain the effective condition number and the conditions in which it is smaller than the original one. However, since FOSI uses two optimizers on orthogonal subspaces, a more relevant measure for improvement is whether the effective condition number of each subspace is smaller than the original one. We show that the condition number of the subspace that is spanned by $\widehat{V}$ is 1 and the condition number of $\widetilde{V}$ is $\lambda_{k+1}/\lambda_{n-\ell}$. Both condition numbers, of $f_1$ and $f_2$, are smaller than the condition number of the original $H$. See Appendices A.4 for proofs and analysis.

### 3.4 Momentum

Momentum accelerates convergence of first-order optimizers and is adapted by many popular optimizers (Qian, 1999; Kingma & Ba, 2014). When using momentum, the descent direction is computed on $\bar{g}$, instead of $g$, where $\bar{g}$ is a linear combination of the current and past gradients. Momentum could also be used by FOSI; however, FOSI and the base optimizer must apply the same linear combination on $g_1$ and $g_2$, which entails $\bar{g} = \bar{g_1} + \bar{g_2}$, to maintain the orthogonality of $f_1$ and $f_2$. We can use $\bar{g}, \bar{g_1}, \bar{g_2}$ in the proof of Lemma 1, instead of $g, g_1, g_2$ and obtain similar results.

### 3.5 Convergence in the Stochastic Setting

We adopt the stochastic setting proposed by Wang et al. (2017). Consider the stochastic optimization problem $\min_\theta f(\theta)$ for $f(\theta) = \mathbb{E}_x[F(\theta, x)]$, where $F : \mathbb{R}^n \times \mathbb{R}^d \to \mathbb{R}$ is twice differentiable w.r.t $\theta$ and $x \in \mathbb{R}^d$ denotes a random variable with distribution $P$. When stochastic optimization is used for DNN training, $f$ is usually approximated by a series of of functions: at each iteration $t$, a batch $b_t$ containing $m_t$ data samples $\{x_1, x_2, \dots, x_{m_t}\}$ is sampled and the function $f^t$ is set as $f^t(\theta) = \frac{1}{m_t} \sum_{i=1}^{m_t} F(\theta, x_i)$. Note that labels, if any, can be added to the data vectors to conform to this model. Adapting FOSI to stochastic DNN training requires a small change to FOSI's algorithm. at the beginning of each iteration, the first action would be to sample a batch $b_t$ and set $f^t$. We can call the ESE procedure with the current $f^t$, or with some predefined $f^i, i \leq t$, as discussed in §3.7.

We now show convergence of FOSI in the common stochastic setting under common Lipschitz smoothness assumptions on $f$ and $F$, and assuming bounded noise level of the stochastic gradient.

**Lemma 2.** *Let BaseOpt be a first-order optimizer that utilizes a positive definite diagonal preconditioner and denote by $\nabla^2 F(\theta, x) = \frac{\partial^2 F}{\partial \theta^2}$ the Hessian of $F$ w.r.t $\theta$. Assuming:*

1. *$f(\theta)$ is $L$-smooth and lower bounded by a real number.*
2. *For every iteration $t$, $\mathbb{E}_{x_t}[\nabla_\theta F(\theta_t, x_t)] = \nabla f(\theta_t)$ and $\mathbb{E}_{x_t}[\|\nabla_\theta F(\theta_t, x_t) - \nabla f(\theta_t)\|^2] \leq \sigma^2$, where $\sigma > 0$, $x_t$ for $t = 1, 2, \dots$ are independent samples, and for a given $t$ the random variable $x_t$ is independent of $\{\theta_i\}_{i=1}^t$.*
3. *There exist a positive constant $z$ s.t. for every $\theta$ and $x$, $\|\nabla^2 F(\theta, x)\| \leq z$ and the diagonal entries of BaseOpt's preconditioner are upper bounded by $z$.*

*Then, for a given $\epsilon \in (0, 1)$, the number of iterations $N$ needed to obtain $\frac{1}{N} \sum_{t=1}^N \mathbb{E}[\|\nabla f(\theta_t)\|^2] \leq \epsilon$ when applying FOSI with BaseOpt is $N = O(\epsilon^{-1/(1-\beta)})$, for step size $\eta$ chosen proportional to $t^{-\beta}$, where $\beta \in (0.5, 1)$ is a constant.*

The proof (in Appendix A.5) works by expressing FOSI in the stochastic quasi-Newton method form used in Theorem 2.8 of Wang et al. (2017), and proving the Theorem's conditions are satisfied.

In the convex, non-stochastic scenario, the convergence rate of the base optimizer becomes the limiting factor, as Newton's method demonstrates a quadratic convergence rate on $f_1$. Therefore, FOSI's convergence rate mirrors that of the base optimizer for $f$, but with improved constants due to the smaller condition number of $f_2$. For instance, the convergence analysis of GD yields $f(\theta_t) - f(\theta^*) \leq \|\theta_0 - \theta^*\|^2/(2\alpha t)$ for $\alpha \leq 1/L$. In the convex case, $L = \lambda_1$ (the maximal eigenvalue of the Hessian). Since FOSI-GD reduces the maximal eigenvalue to $\lambda_{k+1}$, its bound is tighter.

### 3.6 Automatic Learning Rate Scaling

When the base optimizer has a closed-form expression of its optimal learning rate in the quadratic setting that is only dependant on the extreme eigenvalues, FOSI can adjust a tuned learning rate $\eta$ to better suit the condition number of $f_2$. Fortunately, in most cases of optimizers that utilize a diagonal preconditioner, such as GD, Heavy-Ball, and Nesterov, there are such closed-forms (Lessard et al., 2016). Specifically, when applying FOSI with such a base optimizer and given the relevant closed-form expression for the optimal learning rate, the adjusted learning rate at iteration $t$ would be $\eta_2 = \eta(\eta_2^*/\eta^*)$, where $\eta^*$ is the optimal learning rate for the quadratic approximation $\tilde{f}$ and $\eta_2^*$ is the optimal one for $f_2$. FOSI is able to compute this scaling, using the ESE outputs.

The intuition behind this scaling is that the ratio between the optimal learning rates is proportional to the ratio between the condition number of $\tilde{f}$ and that of $f_2$. The full details regarding this scaling technique are in Appendix A.6. Note that $\eta_2^*/\eta^* \geq 1$. In practice, we suggest a more conservative scaling that involves clipping over this scaling factor as follows: $\eta_2 = \eta \min\{\eta_2^*/\eta^*, c\}$ for $c \geq 1$. For $c = 1$, $\eta_2 = \eta$, and for extremely large $c$ ($\infty$), the scaling factor is not clipped.

### 3.7 Error and Overhead

**ESE Approximation Error.** Using Newton's method in non-quadratic settings in conjunction with inexact approximation of Hessian eigenvalues through the ESE procedure increases the risk of divergence. FOSI uses several techniques to address this: scaled Newton's method, extra numerical accuracy inside the ESE procedure, full orthogonalization, and warmup. The details are available in Appendix A.7. In practice, our experiments on a variety of DNNs in §4 demonstrate that FOSI is robust and substantially improves convergence.

**Runtime.** FOSI's runtime differs from that of the base optimizer due to additional computations in each update step and calls to the ESE procedure. For large and complex functions, the latency of the update step of both optimizers, the base optimizer and FOSI, is negligible when compared to the computation of the gradient in each iteration. Furthermore, since each Lanczos iteration is dominated by the Hessian-vector product operation which takes approximately two gradient evaluations, the latency of the ESE procedure can be approximated by $2m\tau$, where $\tau$ is gradient computation latency and $m$ the number of Lanczos iterations (see §3.1). The ESE procedure is called every $T$ iterations, and the parameter $T$ should be set such that FOSI's runtime is at most $\rho$ times the base optimizer runtime, for a user-defined overhead $\rho > 1$. Thus, given the above approximations and assumptions, we can achieve overhead $\rho$ by setting: $T = 2m/(\rho - 1)$. This heuristic helps avoid the need to tune $T$, though FOSI can of course use any $T > 0$. See Appendix A.8 for additional details as well as a more accurate expression for $T$ for functions where additional computations are not negligible in comparison to gradient computations.

**Memory.** FOSI stores $k + \ell$ eigenvectors of size $O(n)$, and temporarily uses $O(mn)$ memory when performing the ESE procedure. In comparison, other second order methods such as K-FAC (Martens & Grosse, 2015) and Shampoo (Gupta et al., 2018) incur $O(\sum_{i \in L} d_i^2 + p_i^2)$ memory overhead, where $d_i$ is the input dimension of layer $i$, $p_i$ the output dimension, and $L$ the total number of layers.

## 4 Evaluation

We first evaluate FOSI's performance on benchmarks tasks including real-world DNNs with standard datasets, for both first- and second-order methods. We then validate our theoretical results by evaluating FOSI on a positive definite (PD) quadratic function with different base optimizers; we explore the effect of the dimension $n$, the eigenspectrum, the learning rate, the base optimizer, and the clipping parameter $c$ on FOSI's performance. We implemented FOSI in Python using the JAX framework (Bradbury et al., 2018) 0.3.25. For experiments, we use an NVIDIA A40 GPU.

### 4.1 Deep Neural Networks

We evaluated FOSI on five DNNs of various sizes using standard datasets, first focusing first-order methods in common use. We execute FOSI with $k = 10$ and $\ell = 0$, since small eigenvalues are usually negative. We set $\alpha = 0.01$, $c = 3$, and $W$ such that warmup is one epoch. $T$ is determined

Table 1: Wall time in seconds to reach target validation accuracy (AC, TL, LR) or loss (LM, AE). The target (in parentheses) is the best one reached by the base optimizer. No single base optimizer is best for all tasks.

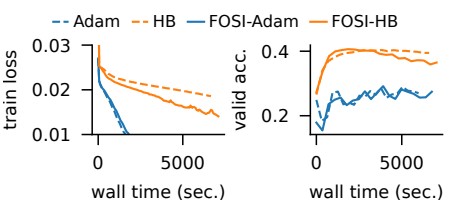

| Task | HB | FOSI-HB | | Adam | FOSI-Adam | |
|------|-----|---------|--------|------|-----------|--------|
| AC | 3822 | **1850** | (40.4%) | 5042 | **3911** | (28.9%) |
| LM | 269 | **207** | (1.71) | 270 | **219** | (1.76) |
| AE | 354 | **267** | (52.46) | 375 | **313** | (51.26) |
| TL | 93 | **53** | (79.1%) | 68 | **33** | (79.0%) |
| LR | 16 | **8** | (92.8%) | **12** | 18 | (92.8%) |

Figure 2: Training AC (MobileNetV1 on AudioSet data). FOSI converges faster than HB and similar to Adam across wall time (left). However, Adam overfits and generalizes poorly as indicated by its low validation accuracy (right).

using the heuristic suggested in § 3.7 aiming at 10% overhead ($\rho = 1.1$), resulting in $T = 800$ for all experiments. The base optimizers compared to FOSI are Heavy-Ball (HB) and Adam; we omit GD (SGD) as it performed worse than HB in most cases. We use the standard learning rate for Adam (0.001), and the best learning rate for HB out of 0.1, 0.01, 0.001, with default momentum parameters $\beta_1 = 0.9, \beta_2 = 0.999$ for Adam and $\beta = 0.9$ for HB. The five evaluated tasks are:

1. **Audio Classification (AC):** Training MobileNetV1 (approximately 4 million parameters) on the AudioSet dataset (Gemmeke et al., 2017). The dataset contains about 20,000 audio files, each 10 seconds long, with 527 classes and multiple labels per file. We converted the audio files into 1-second mel-spectrograms and used them as input images for the DNN. The multi-hot label vector of a segment is the same as the original audio file's label.
2. **Language Model (LM):** Training an RNN-based character-level language model with over 1 million parameters (Hennigan et al., 2020) on the Tiny Shakespeare dataset (Karpathy, 2015). For LM training batches are randomly sampled; there is no defined epoch and we use $W = T$.
3. **Autoencoder (AE):** Training an autoencoder model with roughly 0.5 million parameters on the CIFAR-10 dataset. Implementation is based on Lippe (2022) with latent dimension size 128. We observed that the HB optimizer in this case is sensitive to the learning rate and diverges easily. Therefore we run FOSI with $c = 1$ (prevents learning rate scaling) and $W = T$, which enables extra warmup iterations (number of iteration per epoch is 175).
4. **Transfer Learning (TL):** Transfer learning from ImageNet to CIFAR-10. We start with a pre-trained ResNet-18 on ImageNet2012 and replace the last two layers with a fully-connected layer followed by a Softmax layer. We train the added fully-connected layer (5130 params), while the other layers are frozen (11 million parameters).
5. **Logistic Regression (LR):** Training a multi-class logistic regression model to predict the 10 classes of the MNIST dataset. The model is a neural network with one fully connected layer of 784 input size followed by a Softmax layer of 10 outputs, containing 7850 parameters. The input data is the flattened MNIST images. Since logistic regression is a convex function, the model is also convex w.r.t. the parameters of the network.

Table 1 summarize the experimental results, showing the wall time when reaching a target validation accuracy (for AC, TL, LR tasks) or target validation loss (for LM, AE tasks). The target metric (in parentheses) is the best one reached by the base optimizer. FOSI consistently reaches the target metric faster than the base optimizer (though Adam is faster than FOSI-Adam on LR, FOSI-HB is faster than both). The improvement in FOSI-HB is more significant than in FOSI-Adam, due to FOSI-HB's ability to adapt the learning rate according to the improved condition number of the effective Hessian.

Figure 2 shows the optimizers' learning curves for the AC task. The training loss curves suggests that FOSI significantly improves the performance of HB, but does not help Adam. However, while Adam's training loss reaches zero quickly, it suffers from substantial overfitting and generalizes poorly, as indicated by its accuracy. This supports the idea that there is no single best optimizer for all problems (Zhou et al., 2020). FOSI aims to improve the best optimizer for each specific task. In this case, HB is preferred over Adam as it generalizes much better, and FOSI improves over HB.

Figure 3: Learning curves for minimizing PD quadratic functions $f_H(\theta) = 0.5\theta^T H\theta$ with varying $n$ and $\lambda_1$ values. FOSI converges more than two orders of magnitude faster than its counterparts.

It is important to note that when the base optimizer overfits, FOSI's acceleration of convergence also leads to an earlier overfitting point. This can be observed in the validation accuracy curve: HB begins to overfit near epoch 70 while FOSI begins to overfit at roughly epoch 35. Overall, throughout our experiments, we have not observed FOSI to overfit more than the base optimizer; FOSI always reaches the same or superior validation accuracy as the base optimizer.

**Summary.** FOSI improves convergence of the base optimizer, and is the fastest optimizer for all five tasks. On average, FOSI achieves the same loss as its base optimizer in 78% of the time on the training set and the same accuracy/loss in 75% of the time on the validation set. Thus while the average FOSI iteration takes longer than the base optimizer's, FOSI requires fewer iterations resulting in overall faster convergence. Additional results can be found in Appendix B.1.

## 4.2 Comparison to Second-Order Methods

We compare FOSI to two representative second-order techniques, K-FAC (Martens & Grosse, 2015) and L-BFGS (Liu & Nocedal, 1989); these use a block-diagonal approximation of the Hessian as a preconditioner, subsequently perform its inversion. We repeat the five DNN training experiments and compare the results of both algorithms to FOSI-HB. We use the KFAC-JAX (Botev & Martens, 2022) implementation for K-FAC and the JAXOpt library (Blondel et al., 2021) for L-BFGS.

We used grid search to tune K-FAC's learning rate and momentum, and included K-FAC's *adaptive* as one of the options. We utilized adaptive damping and maintained the default and more precise $T_3$ (interval between two computations of the approximate Fisher inverse matrix) value of 5 after testing larger $T_3$ values and observing no variation in runtime. For tuning L-BFGS hyperparameters, we used line-search for the learning rate, and performed a search for the optimal $L$ (history size) for each task, starting from $L = 10$ (similar to $k$ we used for FOSI) and up to $L = 100$. The hyperparameters were selected based on the lowest validation loss obtained for each experiment.

Overall, we observed that both K-FAC and L-BFGS algorithms have slower runtimes and poorer performance compared to FOSI. They occasionally diverge, can overfit, and rarely achieve the same level of validation accuracy as FOSI. See Appendix B.1 for figures and additional results.

## 4.3 Quadratic Functions

To evaluate FOSI's optimization performance across range of parameters, we use controlled experiments on PD quadratic functions of the form $f_H(\theta) = 0.5\theta^T H\theta$. We use GD, HB, and Adam to minimize $f_H$, as well as FOSI with these base optimizers. We use the default momentum parameters $\beta_1 = 0.9, \beta_2 = 0.999$ for Adam and $\beta = 0.9$ for HB. The learning rate $\eta$ for Adam was set to 0.05 after tuning, for GD to the optimal value $2/(\lambda_1 + \lambda_n)$, and for HB we used $2/(\sqrt{\lambda_1} + \sqrt{\lambda_n})^2$ which is half of the optimal value (due to using constant $\beta$ rather than optimal). FOSI runs with $k = 10$, $\ell = 0$, $\alpha = 1$, and $c = \infty$ (no clipping on the scaling of the GD and HB learning rates, see §3.6).

**Dimensionality and Eigenspectrum.** To study the effect of dimensionality and eigenspectrum on FOSI, we created five $f_H$ functions for each $n \in \{100, 1500\}$ by varying $\lambda_1$ of the Hessian $H$ with $\lambda_1 \in \{5, 10, 20, 50, 200\}$. The other eigenvalues were set to $\lambda_i = 1.5^{-(i-2)}$ and the eigenvectors were extracted from a symmetric matrix whose entries were randomly sampled from $U(0, 1)$.

Figure 3 shows learning curves of the optimizers on functions with $\lambda_1 = 5$ and $\lambda_1 = 200$. Similar results were obtained for other functions. FOSI converges at least two orders of magnitude faster

than its counterparts. In this case, dimensionality has little impact on the performance of different optimizers. For a specific $n$ value, increasing $\lambda_1$ causes the base optimizers to converge to less optimal solutions, but has little impact on FOSI. This is expected for GD and HB, whose learning rate is limited by the inverse of the largest eigenvalue, hence, larger $\lambda_1$ implies slower convergence. FOSI reduces the largest eigenvalue, allowing for larger learning rate that is identical for all functions. Interestingly, this is observed for Adam as well.

**Ill-conditioning and diagonally dominance.** We explore the effect of both the condition number and the diagonally dominance of the function's Hessian on the different optimizers. We use a set of quadratic functions with different condition number and different rotation w.r.t. the coordinate system, which impacts the dominance of the Hessian's diagonal. While all optimizers are negatively affected by large condition number, only Adam is affected by the rotation. FOSI improves over the base optimizer in all cases. The full details and analysis of the results are in Appendix B.2.1

**Learning rate and momentum.** We explored the effect of various learning rates and momentum parameters on the optimizers. We find that FOSI improves over Adam, HB, and GD for all learning rates and momentum (for HB and Adam). The full details of this experiment are in Appendix B.2.2.

## 5 RELATED WORK

Partially second-order optimizers are a group of optimization methods that incorporate some aspects of second-order information in their optimization process. Optimizers that use a diagonal preconditioner (Yao et al., 2021; Jahani et al., 2022; Henriques et al., 2019; Liu et al., 2023), and in fact approximate the Hessian diagonal, suffer when the assumption for diagonally dominance Hessian does not hold (see § 4.3). L-BFGS (Liu & Nocedal, 1989), which uses low-rank approximation of the Hessian, is sensitive to the rank parameter and an incorrect selection can lead to slow convergence or divergence. Additionally, it requires line search in each iteration, slowing down the optimization process further. Recent approaches, such as K-FAC (Martens & Grosse, 2015), Shampoo (Gupta et al., 2018), K-BFGS (Goldfarb et al., 2020), LocoProp (Amid et al., 2022), Eva (Zhang et al., 2023), and Yang et al. (2023) exploit the structure of the network to approximate a block diagonal preconditioner matrix, as an alternative to full second-order methods. However, these techniques approximate the preconditioner directly instead of approximating its inverse, potentially resulting in higher approximation errors and noise sensitivity (Li, 2017). They also exhibit comparable limitations to those of diagonal preconditioners due to neglecting Hessian elements outside the diagonal blocks, such as inter-layer parameter correlations or rotated problems (§4.3). In contrast, by splitting the problem into two subspaces FOSI obtains a full low-rank representation of the Hessian for the first subspace $\widehat{V}$, which captures both the rotation and curvature of the sub-problem $f_1$. This contributes to accuracy and stability of the optimization, particularly as it is based on extreme eigenvalues and vectors that can be approximated more accurately.

Other optimization approaches for stochastic settings involve the use of sub-sampling of $f^i$ functions and constructing an approximation of the Hessian based on the gradients of these functions at each iteration (Roosta-Khorasani & Mahoney, 2019; Xu et al., 2016). However, these methods are limited to functions with only a few thousand parameters. Hessian-free optimization methods (Martens et al., 2010; Martens & Sutskever, 2011; Frantar et al., 2021) rely on conjugate gradient to incorporate second order information, which while more efficient than Lanczos in terms of memory, it still often requires many steps to converge and is more sensitive to noise. Finally, while these works propose a single improved optimizer, FOSI is a meta-optimizer.

## 6 DISCUSSION AND FUTURE WORK

FOSI is a hybrid meta-optimizer that combines a first-order base optimizer with Newton's method to improve the optimization process without additional tuning. Evaluation on real and synthetic tasks demonstrates FOSI improves the wall time to convergence when compared to the base optimizer. Future research will focus on methods for automatic tuning of different parameters of FOSI, such as dynamically adjusting parameters $k$ and $\ell$ according to their impact on the effective condition number. We also plan to investigate the effect of stale spectrum estimation, which could allow running the ESE procedure on the CPU in parallel to the training process on the GPU.

ACKNOWLEDGMENTS

The authors thank the anonymous reviewers for their valuable feedback. The research leading to these results was supported by the Israel Science Foundation (grant No.191/18), the Technion Hiroshi Fujiwara Cyber Security Research Center, the Israel National Cyber Directorate, and the HPI-Technion Research School.

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

# Appendix

## A First and Second-Order Integration

### A.1 The ESE Algorithm

This section describes the ESE algorithm for obtaining the $k$ largest and $\ell$ smallest eigenvalues, as well as their corresponding eigenvectors, of the Hessian $H_t$. The full details of the algorithm are in § 3.1.

ESE first sets the number of Lanczos iterations, $m$, defines the $hvp_t$ operator, and then calls the Lanczos algorithm. After running Lanczos for $m$ iterations, its output is a matrix $U \in \mathbb{R}^{n \times m}$ with orthonormal columns and a tridiagonal real symmetric matrix $T \in \mathbb{R}^{m \times m}$

To extract the approximate eigenvalues and eigenvectors of $A$, let $Q \Lambda Q^T$ be the eigendecomposition of $T$, s.t. $\Lambda$ is a diagonal matrix whose diagonal is the eigenvalues of $T$ sorted from largest to smallest and $Q$'s columns are their corresponding eigenvectors. The approximate $k$ largest and $\ell$ smallest eigenvalues of $A$ are the first $k$ and last $\ell$ elements of $\Lambda$'s diagonal, and their approximate corresponding eigenvectors are the first $k$ and last $\ell$ columns of the matrix product $UQ$.

Algorithm 1 details the ESE procedure.

---

**Algorithm 1** Extreme Spectrum Estimation.

---

**procedure** ESE$(f, \theta_t, k, \ell)$
    $n \leftarrow$ length of $\theta_t$
    $m \leftarrow \max\{4(k + \ell), 2 \ln n\}$
    $hvp_t \leftarrow$ generate $hvp$ operator from $f$ and $\theta_t$.
    $U, T \leftarrow \text{Lanczos}(m, hvp_t)$
    $Q, \Lambda \leftarrow \text{eigendecomposition}(T)$
    $\widehat{\lambda} \leftarrow$ first $k$ and last $\ell$ entries of $\Lambda$'s diagonal
    $\widehat{V} \leftarrow$ first $k$ and last $\ell$ columns of $UQ$
    **return** $\widehat{\lambda}, \widehat{V}$

---

### A.2 The FOSI Algorithm

Algorithm 2 provides the pseudocode for FOSI. The details of the algorithm are in § 3.2.

### A.3 Preconditioner Analysis

We start by proving Lemma 1 from § 3.3, and continue by analysing a special case in which the eigenvectors of $H$ are aligned to the axes of the Euclidean space.

*Proof.* In case we apply FOSI on an optimizer that uses an inverse diagonal preconditioner s.t. $d_b = -\eta \operatorname{diag}(\mathbf{q}) g_2$, then:

$$d_1 = -\alpha \widehat{V} \left( \left( \widehat{V}^T g_1 \right) \odot \mathbf{u} \right) = -\alpha \widehat{V} \left( \left[ \underbrace{\widehat{V}^T \widehat{V}}_{I} \left( \widehat{V}^T g \right) \right] \odot \mathbf{u} \right) = -\alpha \widehat{V} \operatorname{diag}(\mathbf{u}) \widehat{V}^T g,$$

$$d_2 = d_b - \widehat{V} \left( \widehat{V}^T d_b \right) = \left( I - \widehat{V} \widehat{V}^T \right) d_b = -\eta \left( I - \widehat{V} \widehat{V}^T \right) \operatorname{diag}(\mathbf{q}) g_2$$
$$= -\eta \left( I - \widehat{V} \widehat{V}^T \right) \operatorname{diag}(\mathbf{q}) \left( g - \widehat{V} \left( \widehat{V}^T g \right) \right) = -\eta \left( I - \widehat{V} \widehat{V}^T \right) \operatorname{diag}(\mathbf{q}) \left( I - \widehat{V} \widehat{V}^T \right) g.$$

By assigning these forms of $d_1$ and $d_2$ in the update step $\theta_{t+1} = \theta_t + d_1 + d_2$, we obtain that the update step is of the form $\theta_{t+1} = \theta_t - P^{-1} g$ and the inverse preconditioner is:

$$P^{-1} = \alpha \widehat{V} \operatorname{diag}(\mathbf{u}) \widehat{V}^T + \eta \left( I - \widehat{V} \widehat{V}^T \right) \operatorname{diag}(\mathbf{q}) \left( I - \widehat{V} \widehat{V}^T \right). \tag{1}$$

Note that

$$\widehat{V} \operatorname{diag}(\mathbf{u}) \widehat{V}^T = V \operatorname{diag}([\mathbf{u}, \mathbf{0}_{n-k-\ell}]) V^T. \tag{2}$$

---

**Algorithm 2** FOSI Optimizer.

---

**initialization:**
1: BaseOptStep: given gradient, return descent direction.
2: $T$: number of iterations between two ESE runs.
3: $W$: number of warmup iterations before calling ESE.
4: $k, \ell$: parameters for ESE, $1 \leq k + \ell \ll n$.
5: $\alpha$: positive learning rate (scalar).
6: $\mathbf{u} \leftarrow \mathbf{0}, \widehat{V} \leftarrow \mathbf{0}$.
**procedure** UPDATESTEP($\theta, g, \widehat{V}, \mathbf{u}$)
7: $g_1 \leftarrow \widehat{V}\left(\widehat{V}^T g\right), \quad g_2 \leftarrow g - \widehat{V}\left(\widehat{V}^T g\right)$
8: $d_1 \leftarrow -\alpha\widehat{V}\left(\left(\widehat{V}^T g_1\right) \odot \mathbf{u}^T\right)$
9: $d_b \leftarrow$ BaseOptStep($g_2$)
10: $d_2 \leftarrow d_b - \widehat{V}\left(\widehat{V}^T d_b\right)$
11: $\theta \leftarrow \theta + d_1 + d_2$
12: **return** $\theta$
**procedure** OPTIMIZE($f, \theta_0$)
13: $t \leftarrow 0$
14: **while** $\theta_t$ not converged **do**
15: $\quad g_t \leftarrow \nabla f(\theta_t)$
16: $\quad$ **if** $t >= W$ and $(t - W) \mod T = 0$ **then**
17: $\quad\quad \widehat{\boldsymbol{\lambda}}, \widehat{V} \leftarrow$ ESE($f, \theta_t, k, \ell$)
18: $\quad\quad \mathbf{u} \leftarrow 1/|\widehat{\boldsymbol{\lambda}}|$
19: $\quad \theta_{t+1} \leftarrow$ UpdateStep($\theta_t, g_t, \widehat{V}, \mathbf{u}$)
20: $\quad t \leftarrow t + 1$

---

Similarly, and using the fact that $V$ is an orthonormal matrix ($V$ is an orthogonal basis) and hence $VV^T = I$:

$$I - \widehat{V}\widehat{V}^T = VIV^T - V\operatorname{diag}([\mathbf{1}_{k+\ell}, \mathbf{0}_{n-k-\ell}])V^T = V\left(I - \operatorname{diag}([\mathbf{1}_{k+\ell}, \mathbf{0}_{n-k-\ell}])\right)V^T$$
$$= V\operatorname{diag}([\mathbf{0}_{k+\ell}, \mathbf{1}_{n-k-\ell}])V^T. \tag{3}$$

By assigning equation 2 and equation 3 in equation 1 we obtain:

$$P^{-1} = \alpha V\operatorname{diag}([\mathbf{u}, \mathbf{0}_{n-k-\ell}])V^T + \eta V\operatorname{diag}([\mathbf{0}_{k+\ell}, \mathbf{1}_{n-k-\ell}])V^T \operatorname{diag}(\mathbf{q})V\operatorname{diag}([\mathbf{0}_{k+\ell}, \mathbf{1}_{n-k-\ell}])V^T$$
$$= V[\ \alpha\operatorname{diag}([\mathbf{u}, \mathbf{0}_{n-k-\ell}]) + \eta\operatorname{diag}([\mathbf{0}_{k+\ell}, \mathbf{1}_{n-k-\ell}])V^T \operatorname{diag}(\mathbf{q})V\operatorname{diag}([\mathbf{0}_{k+\ell}, \mathbf{1}_{n-k-\ell}])\ ]V^T.$$

This completes the proof of claim 1 of the Lemma.

Note that multiplying a diagonal matrix from the left of another matrix is equivalent to scaling each row of the later by the corresponding diagonal entry of the former, and similarly multiplying a diagonal matrix from from the right has the same effect on columns. Therefore, the matrix

$$B = \operatorname{diag}([\mathbf{0}_{k+\ell}, \mathbf{1}_{n-k-\ell}])V^T \operatorname{diag}(\mathbf{q})V\operatorname{diag}([\mathbf{0}_{k+\ell}, \mathbf{1}_{n-k-\ell}]) \tag{4}$$

is a matrix whose first $k + \ell$ rows and first $k + \ell$ columns are 0.

Denote by $M$ the sub matrix of $B$ that contains the entries $i, j$ s.t. $i, j > k + \ell$. Sine BaseOpt utilizes a positive definite (PD) preconditioner (as stated in the Lemma 1), i.e. $\operatorname{diag}(\mathbf{q}) > 0$, hence $V^T \operatorname{diag}(\mathbf{q})V > 0$, and since $M$ is a trailing principal submatrix of $V^T \operatorname{diag}(\mathbf{q})V$ it is PD (Gentle, 2017, p. 349). $M$ is also symmetric, since $B$ is symmetric. The diagonal matrix $\operatorname{diag}(\mathbf{u})$ is also symmetric and PD. Since a block diagonal matrix is PD if each diagonal block is PD (Gallier et al., 2020) and symmetric if each block is symmetric, the block diagonal matrix

$$\alpha\operatorname{diag}([\mathbf{u}, \mathbf{0}_{n-k-\ell}]) + \eta B = \begin{pmatrix} \alpha\operatorname{diag}(\mathbf{u}) & \mathbf{0} \\ \mathbf{0} & \eta M \end{pmatrix} \tag{5}$$

is PD and symmetric.

Since $V$ has full column and row rank, and the block diagonal matrix equation 5 is PD, the inverse preconditioner

$$P^{-1} = V\begin{pmatrix} \alpha\operatorname{diag}(\mathbf{u}) & \mathbf{0} \\ \mathbf{0} & \eta M \end{pmatrix}V^T$$

is PD (Gentle, 2017, p. 113). It is also symmetric, as equation 5 is symmetric.

Finally, using the fact that the inverse of a PD and symmetric matrix is PD and symmetric, we conclude that the preconditioner $P$ is symmetric and PD. This completes the proof of claim 2 of the Lemma.

We assign the above $P^{-1}$ in $P^{-1}H$ to obtain the effective Hessian:

$$
\begin{aligned}
P^{-1}H &= V \begin{pmatrix} \alpha \operatorname{diag}(\mathbf{u}) & \mathbf{0} \\ \mathbf{0} & \eta M \end{pmatrix} V^T V \operatorname{diag}([\widehat{\boldsymbol{\lambda}}, \widecheck{\boldsymbol{\lambda}}]) V^T \\
&= V \begin{pmatrix} \alpha \operatorname{diag}(\mathbf{u}) \operatorname{diag}(\widehat{\boldsymbol{\lambda}}) & \mathbf{0} \\ \mathbf{0} & \eta M \operatorname{diag}(\widecheck{\boldsymbol{\lambda}}) \end{pmatrix} V^T \\
&= V \begin{pmatrix} \alpha \operatorname{diag}(\mathbf{1}_{k+\ell}) & \mathbf{0} \\ \mathbf{0} & \eta M \operatorname{diag}(\widecheck{\boldsymbol{\lambda}}) \end{pmatrix} V^T.
\end{aligned}
$$

We obtained a partial eigendecomposition of the effective Hessian, which indicates that there are at least $k + \ell$ repetitions of the eigenvalues with value $\alpha$ and its corresponding eigenvectors are $\widehat{V}$'s eigenvectors. This completes the proof of claim 3 of the Lemma. □

In the special case in which the eigenvectors of $H$ are aligned to the axes of the Euclidean space $\mathbb{R}^n$ (i.e. $H$ is diagonal), $V$ is a permutation matrix (has exactly one entry of 1 in each row and each column and 0s elsewhere). Note that $I \operatorname{diag}(\mathbf{q}) I^T$ is an eigendecomposition of $\operatorname{diag}(\mathbf{q})$. Let $\mathcal{P}$ be the permutation of $I$'s columns, such that $\mathcal{P}(I) = V$. Then $V \operatorname{diag}(\mathcal{P}(\mathbf{q})) V^T$ is also an eigendecomposition of $\operatorname{diag}(\mathbf{q})$, i.e:

$$
\operatorname{diag}(\mathbf{q}) = V \operatorname{diag}(\mathcal{P}(\mathbf{q})) V^T.
$$

Therefore,

$$
V^T \operatorname{diag}(\mathbf{q}) V = V^T V \operatorname{diag}(\mathcal{P}(\mathbf{q})) V^T V = \operatorname{diag}(\mathcal{P}(\mathbf{q})),
$$

and $M$ is the $n - k - \ell$ trailing principal submatrix of $\operatorname{diag}(\mathcal{P}(\mathbf{q}))$. In other words, the diagonal of $M$ contains the last $n - k - \ell$ entries of the vector $\mathcal{P}(\mathbf{q})$. By Lemma 1, $\alpha$ is an eigenvalue of $P^{-1}H$ with $k + \ell$ repetitions. From the analysis of $M$ we obtain that the remaining $n - k - \ell$ eigenvalues of $P^{-1}H$ are: $\eta \mathcal{P}(\mathbf{q})_{k+\ell+1} \lambda_{k+1}, \dots, \eta \mathcal{P}(\mathbf{q})_n \lambda_{n-\ell}$. If $\mathbf{q}$ is a good approximation to the Hessian diagonal, then these eigenvalues should be all close to $\eta$ since each $\mathcal{P}(\mathbf{q})_i$ is an approximation to the inverse of $\lambda_{i-\ell}$.

This is an optimal case, since the two optimization problems defined on the two subspaces have condition number of 1, which enable fast convergence. However, this is a very special case and the Hessian in most optimization problems is not diagonal. Moreover, even in this case, which is ideal for a diagonal preconditioner, FOSI provides benefit, since it solves $f_1$ with Newton's method, which obtains an ideal effective condition number over $\widehat{V}$, and provides the base optimizer with $f_2$, which is defined on a smaller subspace $\widecheck{V}$, hence can be viewed as of smaller dimensionality than $f$.

### A.4 Identity Preconditioner

Here we formalize the claims in § 3.3.

**Lemma 3.** *Under the same assumption as in Lemma 1, with BaseOpt that utilizes a scaled identity inverse preconditioner $\eta I$ for some learning rate $\eta > 0$:*

1. *FOSI's resulting inverse preconditioner is $P^{-1} = V \operatorname{diag}([\alpha \mathbf{u}, \eta \mathbf{1}_{n-k-\ell}]) V^T$.*
2. *The preconditioner $P$ is symmetric and PD.*
3. *$\alpha$ is an eigenvalue of the effective Hessian $P^{-1}H$, and $\widehat{V}$'s columns are in the eigenspace of $\alpha$. In addition, the entries of the vector $\eta \widecheck{\boldsymbol{\lambda}}$ are eigenvalues of $P^{-1}H$ and their corresponding eigenvectors are $\widecheck{V}$'s columns.*

*Proof.* The proof immediately follows from Lemma 1 by replacing $\operatorname{diag}(\mathbf{q})$ with $I$.

By replacing $\operatorname{diag}(\mathbf{q})$ with $I$ in equation 4, we obtain

$$
B = \operatorname{diag}([\mathbf{0}_{k+\ell}, \mathbf{1}_{n-k-\ell}]) V^T I V \operatorname{diag}([\mathbf{0}_{k+\ell}, \mathbf{1}_{n-k-\ell}]) = \operatorname{diag}([\mathbf{0}_{k+\ell}, \mathbf{1}_{n-k-\ell}]).
$$

Hence, $M = \mathrm{diag}(\mathbf{1}_{n-k-\ell})$. Assigning this $M$ in $P^{-1}$ given by Lemma 1 obtains:

$$P^{-1} = V \begin{pmatrix} \alpha \, \mathrm{diag}(\mathbf{u}) & \mathbf{0} \\ \mathbf{0} & \eta \, \mathrm{diag}(\mathbf{1}_{n-k-\ell}) \end{pmatrix} V^T = V \, \mathrm{diag}([\alpha\mathbf{u}, \eta \, \mathrm{diag}(\mathbf{1}_{n-k-\ell})]) V^T,$$

which completes the proof of claim 1 of the Lemma.

$P^{-1}$ is diagonal matrix with positive diagonal entries, and therefore symmetric and PD. Its inverse, $P$, is also symmetric and PD, which completes the proof of claim 2 of the Lemma.

We assign the above $P^{-1}$ in $P^{-1}H$ to obtain the effective Hessian:

$$P^{-1}H = V \, \mathrm{diag}([\alpha\mathbf{u}, \eta\mathbf{1}_{n-k-\ell}]) V^T V \, \mathrm{diag}([\widehat{\boldsymbol{\lambda}}, \widecheck{\boldsymbol{\lambda}}]) V^T = V \, \mathrm{diag}([\alpha\mathbf{1}_{k+\ell}, \eta\widecheck{\boldsymbol{\lambda}}]) V^T.$$

We obtained an eigendecomposition of the effective Hessian, which implies there are $k+\ell$ repetitions of the eigenvalues with value $\alpha$ and their corresponding eigenvectors are $\widehat{V}$'s columns, and the entries of $\eta\widecheck{\boldsymbol{\lambda}}$ are eigenvalues and their corresponding eigenvectors are $\widecheck{V}$'s columns. This completes the proof of claim 3 of the Lemma. □

The following Lemma states the conditions in which the effective condition number is smaller than the original condition number when applying FOSI with a base optimizer that utilizes an identity preconditioner.

**Lemma 4.** *Under the same assumption as in Lemma 3, denote the effective condition number induced by BaseOpt by $\kappa$ and the effective condition number induced by FOSI using BaseOpt by $\tilde{\kappa}$. Then, $\tilde{\kappa} \leq \kappa$ in the following cases:*

1. $\alpha < \eta\lambda_{n-\ell}$ and $\frac{\eta\lambda_{k+1}}{\alpha} \leq \frac{\lambda_1}{\lambda_n}$

2. $\eta\lambda_{n-\ell} \leq \alpha \leq \eta\lambda_{k+1}$

3. $\eta\lambda_{k+1} < \alpha$ and $\frac{\alpha}{\eta\lambda_{n-\ell}} \leq \frac{\lambda_1}{\lambda_n}$

*Proof.* The identity preconditioner does not affect the condition number, so we have $\kappa = \lambda_1/\lambda_n$. As stated in claim 2 of Lemma 3, when using FOSI the distinct eigenvalues of the effective Hessian are $\alpha, \eta\lambda_{k+1}, \dots, \eta\lambda_{n-\ell}$. There are now three distinct ranges for $\alpha$ which affect $\tilde{\kappa}$:

1. $\alpha < \eta\lambda_{n-\ell}$. In this case, the smallest eigenvalue of $P^{-1}H$ is $\alpha$ and the largest is $\eta\lambda_{k+1}$, which leads to $\tilde{\kappa} = \eta\lambda_{k+1}/\alpha$; therefore, $\tilde{\kappa} \leq \kappa \iff \frac{\eta\lambda_{k+1}}{\alpha} \leq \frac{\lambda_1}{\lambda_n}$.

2. $\eta\lambda_{n-\ell} \leq \alpha \leq \eta\lambda_{k+1}$. In this case, the smallest eigenvalue of $P^{-1}H$ is $\eta\lambda_{n-\ell}$ and the largest is $\eta\lambda_{k+1}$; therefore, $\tilde{\kappa} = \frac{\lambda_{k+1}}{\lambda_{n-\ell}}$ and $\tilde{\kappa} \leq \kappa \iff \frac{\lambda_{k+1}}{\lambda_{n-\ell}} \leq \frac{\lambda_1}{\lambda_n}$. Sine $\frac{\lambda_{k+1}}{\lambda_{n-\ell}} < \frac{\lambda_1}{\lambda_n}$ is true then $\tilde{\kappa} < \kappa$.

3. $\eta\lambda_{k+1} < \alpha$. In this case the smallest eigenvalue of $P^{-1}H$ is $\eta\lambda_{n-\ell}$ and the largest is $\alpha$; therefore, $\tilde{\kappa} = \frac{\alpha}{\eta\lambda_{n-\ell}}$ and $\tilde{\kappa} \leq \kappa \iff \frac{\alpha}{\eta\lambda_{n-\ell}} \leq \frac{\lambda_1}{\lambda_n}$.

□

While Lemma 4 provides the conditions in which FOSI improves the condition number, as discussed in § 3.3, FOSI is able to accelerate the convergence of the optimization process even when it does not improve the condition number.

To show this phenomenon, we use GD and FOSI with GD as a base optimizer to optimize the quadratic function $f(\theta) = 0.5\theta^T H\theta, \theta \in \mathbb{R}^{100}$. We draw a random orthonormal basis for $f$'s Hessian, $H$, and set its eigenvalues as follows: $\lambda_1, \dots, \lambda_{10}$ are equally spaced in the range $[9, 10]$ and $\lambda_{11}, \dots, \lambda_{100}$ are equally spaced in the range $[0.01, 0.1]$. We used the learning rate $\eta = 0.001$ and run FOSI with $k = 9, \ell = 0, \alpha = 1$. For this setting we have $\lambda_1 = 10, \lambda_{10} = 9, \lambda_{100} = 0.01$ and none of the conditions in Lemma 4 is satisfied. Since $\eta\lambda_{10} < 1$, the only candidate condition in Lemma 4 is condition (3), however, in this case FOSI's effective condition number is $1/(\eta\lambda_{100}) = 100000$, which is much larger than the original condition number of the problem, which is $\lambda_1/\lambda_{100} = 1000$. However, as shown in Figure 4, FOSI converges much faster then GD.

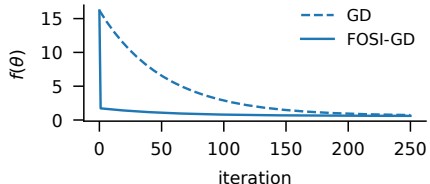

Figure 4: Learning curves of GD and FOSI for the minimization of the quadratic function $f(\theta) = 0.5\theta^T H\theta$, with $\theta \in \mathbb{R}^{100}$. $H$'s eigenvectors are a random orthogonal basis, $\eta = 0.001, \lambda_1 = 10, \lambda_{10} = 9, \lambda_n = 0.01$, and FOSI runs with $k = 9, \ell = 0, \alpha = 1$. While FOSI's effective condition number is larger than the original one, it converges much faster than the base optimizer.

This example emphasises our claim that the condition numbers to look at are those of $\widehat{V}$ and $\widecheck{V}$, which are smaller than the original one, and not the condition number of the entire Hessian.

### A.5 Convergence Guarantees in the Stochastic Setting

Our proof of Lemma 2 relies on applying Theorem 2.8 from Wang et al. (2017) to FOSI. For clarity, we restate their theorem with our notations:

**Theorem 5** (Theorem 2.8, Wang et al. (2017)). *Suppose that the following assumptions hold for $\{\theta_t\}$ generated by a stochastic quasi-Newton (SQN) method with batch size $m_t = m$ for all $t$:*

> (i) *$f$ is continuously differentiable, $f(\theta)$ is lower bounded by a real number $f^{low}$ for any $\theta$, and $\nabla f$ is globally Lipschitz continuous with Lipschitz constant $L$.*
> (ii) *For every iteration $t$, $\mathbb{E}_{x_t}[\nabla_\theta F(\theta_t, x_t)] = \nabla f(\theta_t)$ and $\mathbb{E}_{x_t}[\|\nabla_\theta F(\theta_t, x_t) - \nabla f(\theta_t)\|^2] \leq \sigma^2$, where $\sigma > 0$, $x_t$ for $t = 1, 2, \dots$ are independent samples, and for a given $t$ the random variable $x_t$ is independent of $\{\theta_i\}_{i=1}^t$.*
> (iii) *There exist two positive constants, $\underline{z}, \bar{z}$, such that $\underline{z}I \preceq P_t^{-1} \preceq \bar{z}I$ for all $t$.*
> (iv) *For any $t \geq 2$, the random variable $P_t^{-1}$ depends only on $b_1, b_2, \dots, b_{t-1}$ (the random batch sampling in the $t - 1$ previous iterations).*

*We also assume that $\eta_t$ is chosen as $\eta_t = \frac{\underline{z}}{L\bar{z}^2}t^{-\beta}$ with constant $\beta \in (0.5, 1)$. Then, for a given $\epsilon \in (0, 1)$, the number of iterations $N$ needed to obtain $\frac{1}{N}\sum_{t=1}^N \mathbb{E}\left[\|\nabla f(\theta_t)\|^2\right] \leq \epsilon$ is $N = O\left(\epsilon^{-\frac{1}{1-\beta}}\right)$.*

We now prove Lemma 2.

*Proof.* First, we need to bring FOSI's inverse preconditioner $P^{-1}$ to the standard SQN form stated in Wang et al. (2017), and then prove that all the assumption of Theorem 2.8 are satisfied.

To bring FOSI's $P^{-1}$ from Lemma 2 to the standard SQN form, with update step $\theta_{t+1} = \theta_t - \eta_t P_t^{-1} g_t$, let $\alpha = \eta = -\eta_t$. After extracting $-\eta_t$, $P^{-1}$ is then given by

$$P^{-1} = V \begin{pmatrix} \text{diag}(\mathbf{u}) & \mathbf{0} \\ \mathbf{0} & M \end{pmatrix} V^T,$$

where $M$ is the trailing $n - k - \ell$ principal submatrix of $V^T \text{diag}(\mathbf{q})V$ .

Theorem 2.8 is comprised of four assumptions, where its first two assumption, (i) and (ii), are satisfied by the first two assumptions in Lemma 2. Assumption (iv) requires that for each iteration $t$, FOSI's inverse preconditioner $P^{-1}$ depends only on $b_1, b_2, \dots, b_{t-1}$. This could be easily satisfied by ensuring that the ESE procedure is called with $f^i$ for $i < t$ and that BaseOptStep is called after the gradient step (switching the 4th and 5th steps in the UpdateStep() procedure), while using $d_b$ from the last iteration in the update step. This has no impact on our analysis of FOSI in § 3.3.

To establish assumption (iii) we examine $P^{-1}$ structure. Given that the Hessian is symmetric PSD, it follows that for every $\theta$ and $x$, the norm of the Hessian $\nabla^2 F(\theta, x)$ is equivalent to its largest absolute eigenvalue. Therefore, from assumption 3 that $\left\|\nabla^2 F(\theta, x)\right\| \leq z$, we have that the largest absolute eigenvalue is bounded above by $z$. Since $f^t(\theta) = \frac{1}{m}\sum_{i=1}^m F(\theta, x_i)$, then the largest absolute

eigenvalue of $\nabla^2 f^t(\theta_t)$ is also bounded by $z$. In addition, it should be noted that the ESE procedure provides eigenvalue estimates that are within bounds of the true extreme eigenvalues of the Hessian of $f^t$ (Dorsselaer et al., 2001). Thus, it follows that each entry of $|\widehat{\boldsymbol{\lambda}}|$ is bounded above by $z$, which in turn implies that each entry of $\mathbf{u}$ is bounded below by $1/z$. Moreover, the entries of $\mathbf{u}$ are also bounded above by $1/\epsilon$ for $0 < \epsilon < 1$ ($\epsilon$ is added to entries of $|\widehat{\boldsymbol{\lambda}}|$ that are smaller than $\epsilon$).

Given that BaseOpt utilizes a PD preconditioner (assumption 3), the entries of $\mathbf{q}$ are upper bounded by some positive constant $1/\epsilon$ (assuming w.l.o.g that this is the same constant that upper bounds $\mathbf{u}$). Moreover, given that the eigenvalues of BaseOpt's preconditioner are bounded from above by $z$, the values of $\mathbf{q}$ are lower bounded by $1/z$. Since the matrix $M$ is a trailing principal submatrix of $V^T \operatorname{diag}(\mathbf{q})V$, its eigenvalues are bounded by the eigenvalues of $V^T \operatorname{diag}(\mathbf{q})V$ (*eigenvalue interlacing theorem*), which are simply the entries of $\mathbf{q}$.

Finally, since $V$ is a rotation matrix, a multiplication from the left by $V$ and from the right by $V^T$ has no impact on the eigenvalues, which implies that $P^{-1}$'s eigenvalues are the entries of $\mathbf{u}$ and the eigenvalues of $M$. Hence, for every iteration $t$, $P^{-1}$'s eigenvalues are lower bounded by $\underline{z} = 1/z$ and upper bounded by $\bar{z} = 1/\epsilon$.

After establishing assumptions (i)–(iv), we complete the proof by applying Theorem 2.8 from Wang et al. (2017). □

## A.6 Automatic Learning Rate Scaling

This section provides details regarding the automatic learning rate scaling technique, presented in §3.6.

Let the base optimizer, BaseOpt, be an optimizer with a closed-form expression of its optimal learning rate in the quadratic setting, $\eta$ be a tuned learning rate for BaseOpt over $f$, and $\eta^*$ be the optimal learning rate of BaseOpt over a quadratic approximation $\tilde{f}$ of $f$ at iteration $t$. In general, $\eta^*$ is not known since first-order optimizers do not evaluate the extreme eigenvalues. Implicitly, $\eta$ is a scaled version of $\eta^*$, i.e., $\eta = s\eta^*$ for some unknown positive scaling factor $s$, usually $s < 1$.

FOSI creates a quadratic subproblem, $f_2$, with a lower condition number compared to $\tilde{f}$ and solves it using BaseOpt. Therefore, we propose using $\eta_2 = s\eta_2^*$, a scaled version of the optimal learning rate of $f_2$ with the same scaling factor $s$ of $\eta$, instead of simply using $\eta$. the ESE procedure provides $\lambda_1, \lambda_n, \lambda_k, \lambda_{n-\ell+1}$, which allows FOSI to automatically adjust $\eta$ to $\eta_2$, given the relevant closed-form expression for the optimal learning rate. Specifically, $\eta_2 = \eta(\eta_2^*/\eta^*)$, with $\eta^*$ obtainable from $\lambda_1$ and $\lambda_n$, and an approximate value for $\eta_2^*$ obtainable from $\lambda_k$ and $\lambda n - \ell + 1$.

## A.7 ESE Approximation Error

Using Newton's method in non-quadratic settings within a subspace obtained from the ESE procedure increases the likelihood of divergence due to inaccuracies in the direction of the steps taken. These inaccuracies stem from the imprecise approximation of Hessian eigenvalues and eigenvectors through the ESE procedure.

To mitigate this, we employ a scaled Newton's method, with a learning rate of $0 < \alpha \leq 1$, for function $f_1$, as an alternative to the traditional Newton's method which enforces $\alpha = 1$.

We also avoid issues of numerical accuracy in the ESE procedure by performing full orthogonalization w.r.t all previous vectors in each Lanczos iteration (Meurant & Strakoš, 2006). Moreover, we use float64 for the ESE procedure computations (only); we retain the original precision for training, storing parameters, and other computations.

To mitigate Lanczos divergence caused by a plateau-like initial point (Orvieto et al., 2022), we use warmup iterations before the first ESE call.

Finally, using ESE in a stochastic setting could theoretically result in unsuitable $\widehat{V}$ and $\widehat{\boldsymbol{\lambda}}$ when called with $f^i$ which misrepresent $f$. Techniques to address this include using larger batch size only for the ESE procedure, or averaging the results obtained from different $f^i$s. We leave the investigation of such techniques to future work.

In practice, our experiments on a variety of DNNs in §4, using an arbitrary $f^j$ on ESE calls, demonstrate that FOSI is robust and substantially improves convergence.

### A.8   RUNTIME ANALYSIS

FOSI's runtime differs from that of the base optimizer due to additional computations in each update step and its calls to the ESE procedure.

Let $\tau_1$ be the average latency per iteration of the base optimizers, $\tau_2$ be the average latency per iteration of FOSI that does not include a call to the ESE procedure (as if $T = \infty$), and $\tau_3$ be the average latency of the ESE procedure. Given that the base optimizer and FOSI are run for $T$ iterations, the latency of the base optimizer is $T\tau_1$, and that of FOSI is $T\tau_2 + \tau_3$, as the ESE procedure is called once every $T$ iterations. The parameter $T$ impacts FOSI's runtime relative to the base optimizer. A small $T$ may result in faster convergence in terms of iterations, since $\widehat{V}$ and $\widehat{\lambda}$ are more accurate; however, it also implies longer runtime.[1] On the other hand, using a large $T$ may result in divergence due to inaccurate estimates of $\widehat{V}$ and $\widehat{\lambda}$. Since the improvement in convergence rate, for any given $T$, is not known in advance, the parameter $T$ should be set such that FOSI's runtime it at most $\rho$ times the base optimizer runtime, for a user define $\rho > 1$. To ensure that, we require $\rho T\tau_1 = T\tau_2 + \tau_3$, which implies[2]

$$T = \tau_3/(\rho\tau_1 - \tau_2). \tag{6}$$

The average latency of FOSI's extra computations in an update step (lines 7, 8, and 10 in Algorithm 2), denoted by $\tau_2 - \tau_1$, includes three matrix-vector products and some vector additions, which have a computational complexity of $O(n(k + \ell))$. For large and complex functions, the latency of these extra computations is negligible when compared to the computation of the gradient (line 18 in Algorithm 2)[3], thus leading to the approximation of $\tau_1 \approx \tau_2$. Furthermore, $\tau_3$ can be approximated by $2m\tau_1$, where $m = \max\{4(k + \ell), 2\ln n\}$ is the number of Lanczos iterations, since each Lanczos iteration is dominated by the Hessian-vector product operation which takes approximately two gradient evaluations (see § 3.1). By incorporating these approximations into equation (6), we can derive a formula for $T$ which does not require any measurements:

$$T = 2m/(\rho - 1).$$

Note that this formula is not accurate for small or simple functions, where the gradient can be computed quickly, and the additional computations are not negligible in comparison. In such cases, $\tau_1$, $\tau_2$, and $\tau_3$ can be evaluated by running a small number of iterations, and $T$ can be computed using equation (6) based on these evaluations.

## B   EVALUATION

### B.1   DEEP NEURAL NETWORKS

Figure 5 shows the learning curves of FOSI and the base optimizers for different DNN training tasks: (1) training logistic regression model on the MNIST dataset, (2) training autoencoder on the CIFAR-10 dataset, (3) transfer learning task in which we train the last layer of trained ResNet-18 on the CIFAR-10 dataset, and (4) training character-level language model with a recurrent network on the Tiny Shakespeare dataset. FOSI improves over the base optimizers in most cases. While FOSI improvement over Adam is less significant than its improvements over Heavy-Ball, there are tasks for which Heavy-Ball performs better than Adam since it generalizes better.

---

[1]We do note, however, that in some settings, such as distributed settings in which network bandwidth is limited, using fewer iterations is preferred, even at the cost of additional runtime per iteration. In future work we plan to run the ESE procedure on the CPU in the background in the effort of saving this extra runtime altogether.

[2]It should be noted that this calculation does not take into account the extra evaluation steps during the training process, which has identical runtime with and without FOSI; hence, FOSI's actual runtime is even closer to that of the base optimizer.

[3]For DNNs, gradient computation can be parallelized over the samples in a batch, however, it must be executed serially for each individual sample. In contrast, operations such as matrix-vector multiplication and vector addition can be efficiently parallelized.

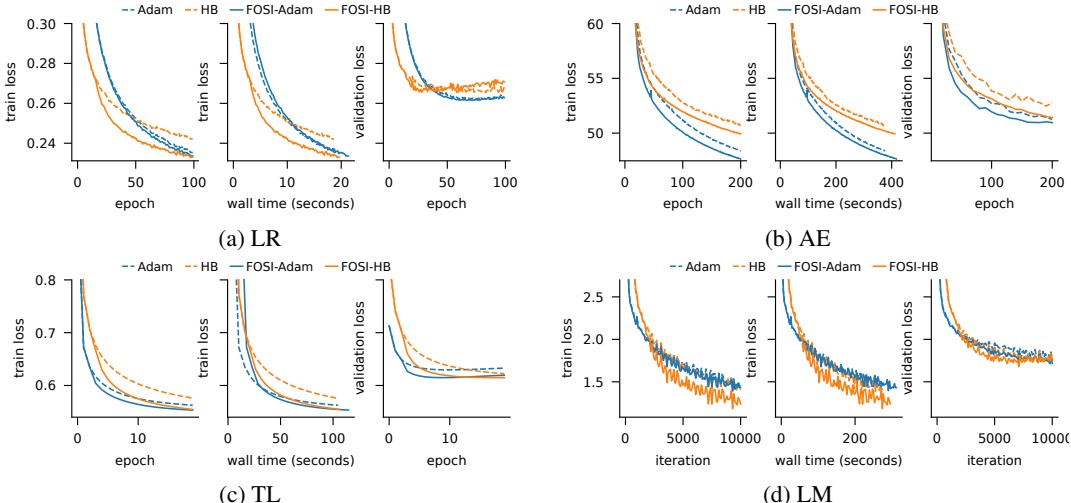

Figure 5: Learning curves of different optimizers for different DNN training tasks. In most cases FOSI obtains faster convergence than the base optimizers across epochs (left) and across wall time (middle). Since FOSI accelerates convergence, it also leads to an earlier overfitting point when the base optimizer has a tendency to overfit, as can be observed in the LR validation loss (right).

Table 2: Comparison of wall time (in seconds) for each base optimizer and FOSI at the same train loss, which is the minimal train loss of the base Optimizer. A lower wall time is preferable as it indicates the optimizer reaches the best loss at a faster rate.

| Task | HB | FOSI-HB | Adam | FOSI-Adam |
|------|------|---------|------|-----------|
| AC | 6845 | **3599** | **6042** | 6825 |
| LM | 255 | **177** | 255 | **233** |
| AE | 372 | **322** | 375 | **338** |
| TL | 103 | **58** | 104 | **71** |
| LR | 18 | **10** | **19** | 19 |

Table 2 shows the time it takes for both the base optimizer and FOSI to reach the same train loss, which is the lowest train loss of the base optimizer. On average, FOSI achieves the same loss over the training set in 78% of the wall time compared to the base optimizers.

Figure 6 shows the learning curves of FOSI-HB, K-FAC and L-BFGS. In all cases, FOSI converges faster and to a lower validation loss than K-FAC and L-BFGS. Specifically:

- LR: K-FAC converges quickly but overfits dramatically, resulting in much higher validation loss than FOSI. L-BFGS converges much more slowly than the other approaches and to a much higher validation loss.
- TL: Both K-FAC and L-BFGS converge slower than FOSI and result in higher validation loss.
- AE: K-FAC converges quickly but is noisy and leads to a large validation loss (52.1 compared to 51.4 for FOSI), while L-BFGS diverges quickly after the first epoch, even with large values of *L*.
- LM: we could not get the K-FAC implementation to work on this RNN model (this is a known issue with K-FAC and RNN (Martens et al., 2018)). L-BFGS converges more slowly, and to a much higher loss.
- AC: K-FAC converges slower than FOSI and shows substantial overfitting, while L-BFGS does not converge.

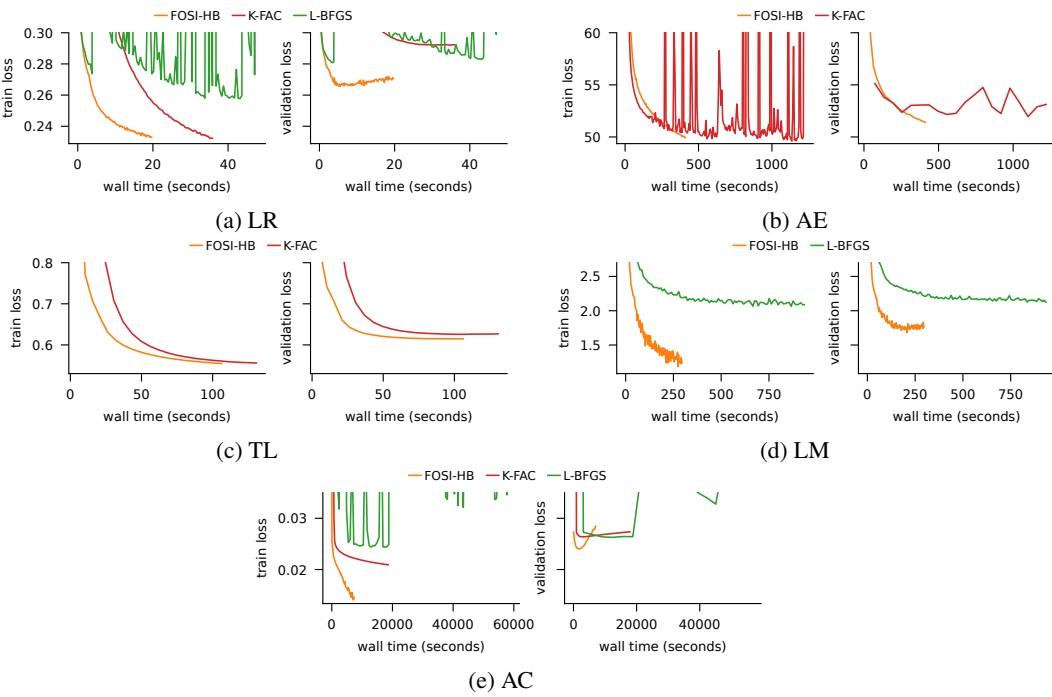

Figure 6: Learning curves of FOSI-BH, K-FAC, and L-BFGS for the DNN training tasks AE, TL, LM, and AC. In the TL figure (top-right), L-BFGS (L=40) was omitted due to its poor performance, resulting in a significantly larger loss than other optimizers and 8x slower wall time. Similarly, in the AE figure (top-left), L-BFGS (L¡=100) was excluded due to its divergence in the first epoch. In the LM figure (bottom-left), K-FAC was omitted due to integration issues with RNN. Across all tasks, FOSI demonstrated faster convergence to a lower validation loss compared to K-FAC and L-BFGS.

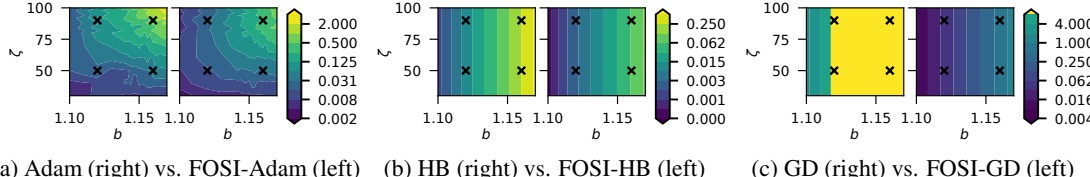

(a) Adam (right) vs. FOSI-Adam (left)  (b) HB (right) vs. FOSI-HB (left)  (c) GD (right) vs. FOSI-GD (left)

Figure 7: Each $(b, \zeta)$ combination in each sub figure is the value of different $f_{b,\zeta}$ after 200 iterations of the optimizer. FOSI improves over the base optimizer for every function. Learning curves for four functions, indicated by black x markers, can be found in Figure 8.

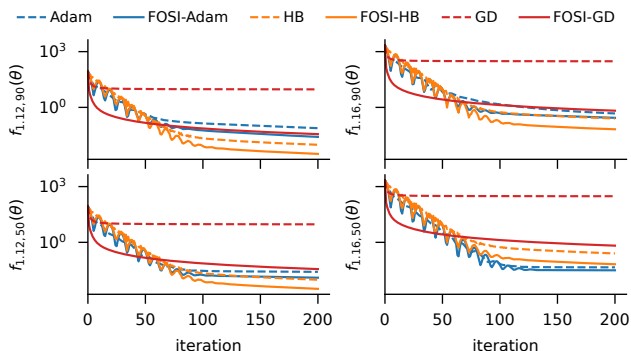

Figure 8: Learning curves of four specific $f_{b,\zeta}$ functions. Each x mark in Figure 7 is the final value of the corresponding learning curve here, i.e. the value of $f$ at iteration 200.

## B.2 Quadratic Functions

### B.2.1 Dimensionality and Eigenspectrum

Here we provide the full details regarding the experiment in § 4.3, where we explore the effect of both the condition number and the diagonally dominance of the function's Hessian on the different optimizers. To do so, we define a set of functions $f_{b,\zeta}(\theta) = 0.5\theta^T H_{b,\zeta}\theta$ for $b \in \{1.1, \ldots, 1.17\}$ and $\zeta \in \{0, \ldots, 100\}$, $\theta \in \mathbb{R}^{100}$, where $b$ and $\zeta$ are parameters that define the Hessian $H_{b,\zeta}$. The parameter $b$ determines the eigenvalues of $H_{b,\zeta}$: $\forall i \in \{1, \ldots, 100\}$ $\lambda_i = 0.001b^i$. The parameter $\zeta$ determines the number of rows in $H_{b,\zeta}$ that are not dominated by the diagonal element, i.e., the part of $H_{b,\zeta}$ which is not diagonally dominant.

To construct $H_{b,\zeta}$, we start from a diagonal matrix whose diagonal contains the eigenvalues according to $b$. We then replace a square block on the diagonal of this matrix with a PD square block of dimensions $\zeta \times \zeta$ whose eigenvalues are taken from the original block diagonal and its eigenvectors are some random orthogonal basis. The result is a symmetric PD block diagonal $H_{b,\zeta}$ with one block of size $\zeta \times \zeta$ and another diagonal block, and the eigenvalues are set by $b$.

An important observation is, that for a specific $b$ value, $b_1$, and two different $\zeta$ values, $\zeta_1$ and $\zeta_2$, the Hessians $H_{b_1,\zeta_1}$ and $H_{b_2,\zeta_2}$ share the same eigenvalues and their eigenvectors are differ by a simple rotation. The starting point $\theta_0$ in all the experiments is the same and it is rotated by a rotation matrix that is $H_{b,\zeta}$'s eigenvectors. As a result, for all the experiments with the same $b$, the starting values $f_{b,\zeta}(\theta_0)$ are identical.

Figure 7 shows $f_{b,\zeta}$ at the optimal point after 200 iterations of the optimizers for different $b$ and $\zeta$ values. For a specific $b$ value, rotations of the coordinate system (changes in $\zeta$) have no impact on GD and HB, as seen by the vertical lines with the same value for different $\zeta$ values. Their performance deteriorates for larger $b$ values (more ill-conditioned problems). When applying FOSI, the new maximal eigenvalues of two functions with similar $\zeta$ and different $b$ are still differ by an order of magnitude, which leads to the differences in FOSI's performance along the $b$ axis. Adam's performance is negatively affected for large $b$ and $\zeta$ values. FOSI improves over the base optimizer in all cases.

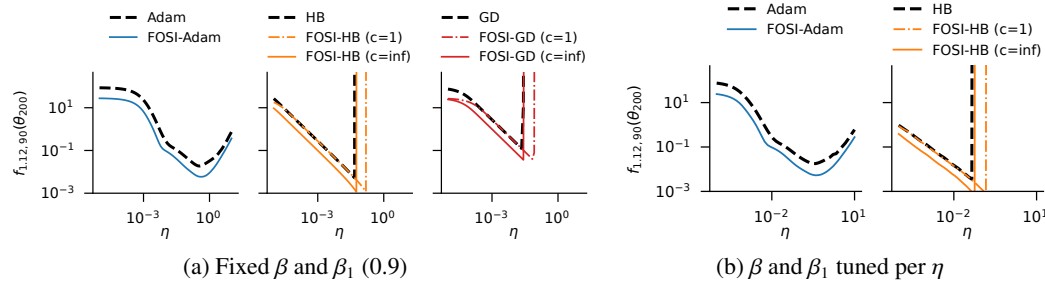

(a) Fixed $\beta$ and $\beta_1$ (0.9)  (b) $\beta$ and $\beta_1$ tuned per $\eta$

Figure 9: $f_{1.12,90}$ after 200 iterations with different learning rates. Each curve is for a different optimizer, while each point in the curve is the final $f$ value after 200 iterations with a specific learning rate. Left: using a fixed momentum value of 0.9, $\beta$ for HB and $\beta_1$ for Adam. Right: the best momentum $\in [0.7, 1)$ for each $\eta$. FOSI improves over the base optimizer even when using the optimal hyperparameter set.

Figure 8 shows the learning curves of the optimizers for four specific $f_{b,\zeta}$ functions: $f_{1.12,50}, f_{1.12,90}, f_{1.16,50}, f_{1.16,90}$. The black x marks in each sub figure of Figure 7 are the last value of these learning curves. For both functions with $b = 1.12$ the learning curves of GD and Heavy-Ball (as well as FOSI with these base optimizers) are identical, as they are only differ by a rotation, and similarly for $b = 1.16$. However, for the same $\zeta$, these optimizers convergence is much slower for larges $b$ value. FOSI implicitly reduces the maximal eigenvalue in both functions, but the new two maximal eigenvalues still differ by an order of magnitude, which leads to the differences in FOSI's performance (as opposed to the first experiment on quadratic functions). Adam is negatively impacted when $\zeta$ is increased. In this experiment. For smaller $\zeta$ values its performance is not impacted by the change in $b$ and it is able to converge to the same value even for functions with larger curvature.

### B.2.2 LEARNING RATE AND MOMENTUM

In the last experiment on quadratic functions, we use each optimizer to optimize the function $f_{1.12,90}$ multiple times, with different learning rates $\eta \in \{1e - 5, 10\}$. FOSI-HB and FOSI-GD were run with both $c = 1$ (no scaling) and $c = \infty$ (no clipping). We repeated the experiment twice. In the first version we used a fixed momentum parameter 0.9 ($\beta$ for HB and $\beta_1$ for Adam). In the second version we find the best momentum parameter $\in [0.7, 1)$ for each $\eta$.

Figure 9 shows the results after 200 iterations for every optimizer and learning rate $\eta$. FOSI improves over Adam for all learning rates. For GD and HB, with $c = 1$, FOSI expands the range of $\eta$ values for convergence, changes the optimal $\eta$, and leads to superior results after the same number of iterations. With $c = \infty$, FOSI improves over the base optimizer for all $\eta$ values, but the range of $\eta$ values for convergence stays similar. Moreover, FOSI's improvement over the base optimizer when using the optimal set of hyperparameters is similar to the improvement for a fixed momentum parameters. Similar trends were observed when repeating the experiment for other $f_{b,\zeta}$ functions.

