# OpenReview forum: "FOSI: Hybrid First and Second Order Optimization"
_ICLR.cc/2024/Conference — ICLR 2024 poster_

### Official Review · Reviewer_S4Ex · 2023-10-28

**Soundness:** 2 fair
**Presentation:** 2 fair
**Contribution:** 3 good
**Rating:** 6
**Confidence:** 2

**Summary:**

his paper provides a novel algorithm to accelerate the first-order optimizers by incorporating second-order information. The proposed method is well-motivated and the experimental results show the efficiency of it.

**Strengths:**

The method is novel which incorporate the descent direction of first-order methods (base optimizer) with second-order information. The empirical study is comprehensive and show the efficiency of the proposed methods. This paper is an interesting attempt on accelerating the first-order methods by using second-order information.

**Weaknesses:**

The method requires to compute the extreme eigenvalues and vectors of Hessian by lanczos algorithm, it will raises much more computational cost per iteration than the first-order methods. Additionally, it is unknown how to choose $k$ and $l$ in the proposed algorithm. Neither the theoretical analysis and empirical study miss the part on evaluating how $k$ and $l$ affect the behavior of the methods. Besides, the parameters are too much, not only $k$, $l$ need to be chosen, Algorithm 2 also requires the parameter for learning rates $\alpha$, the learning rate for the base optimizer.

**Questions:**

1. There are some optimizers which also use the hybrid directions of first and second-order methods, or partial information of the Hessian. The author may need to compare and discuss them [1 ,2].

2. Can the authors provide some discussion and numerical evaluation on how to choose $k$ and $l$.


**Reference**

[1]. Zhang C, Ge D, Jiang B, et al. DRSOM: A Dimension Reduced Second-Order Method and Preliminary Analyses[J]. arXiv preprint arXiv:2208.00208, 2022.

[2]. Liu H, Li Z, Hall D, et al. Sophia: A Scalable Stochastic Second-order Optimizer for Language Model Pre-training[J]. arXiv preprint arXiv:2305.14342, 2023.

---

> ### Author Response · Authors · 2023-11-14
>
> Thank you for this review. We appreciate the feedback and will address each comment and question in the order they were presented.
>
> **W1:** While the average FOSI iteration takes longer, FOSI requires much fewer iterations to reach the same loss compared to the base optimizer. This can be visualized in Fig 5 in the supplementary material. FOSI's ability to use fewer steps is due to its utilization of additional second-order information in each iteration, thereby facilitating more precise descent steps. We stress that the crucial factor is the time to convergence, not the number of internal steps (iterations).
>
> **W2 + Q2:**
> Since our focus was for FOSI as a drop-in replacement without additional tuning, we did not tune $k$ or $\ell$. Instead, for optimizing non-convex losses like DNNs, we recommend setting $k$ to a small number (less than the number of classes), and  $\ell$ to 0. This is supported by the common observation that the number of extreme eigenvalues is generally small, on the order of the number of classes (see for example reference [R1] below), and small eigenvalues predominantly exhibit negative values in real DNN settings. It is worth noting that even if $k$ is lower than the actual number of outlier eigenvalues, the distance between these large eigenvalues can potentially be large, hence the effect on the condition number is still considerable.
> Similarly, $\alpha$ can simply be set to a small conservative constant such as 0.01, since the bottleneck for convergence is the base optimizer used for $f_2$, not the Newton method optimizer for $f_1$.
>
> In practice, as supported by our experiments, it is perfectly fine to simply set $k=10$, $\ell=0$ and $\alpha=0.01$ across a wide range of tasks and architectures.
>
> In our future research, we aim to develop innovative techniques that dynamically adjust $k$ and $\ell$ based on observed improvements in the effective condition number. This approach will enable us to strike a balance between computational cost and optimization performance. However, we acknowledge that this research extends beyond the scope of the current paper.
>
> **Q1:** Concerning Sophia, we emphasize that, despite the paper's title containing "Second-order Optimizer," Sophia bears a close resemblance to Adam, as both approximate the Hessian diagonal. The distinction lies in Adam's utilization of the second moment of the gradient and sqrt for a conservative approximation, whereas Sophia employs the gradient of predictions and clipping. As such, we can expect Sophia to suffer from similar issues as Adam for non-diagonal Hessians (see Section 4.3 and Section B.2.1). We will add this discussion to the paper.
>
> DRSOM is a first-order method, specifically an approach for determining the most accurate step size for gradient methods incorporating Polyak's momentum. To our knowledge, DRSOM has not been applied to DNNs, but for problems with $n<1000$ dimensions and small datasets. We have been unable to find a Python implementation to support a direct comparison for (our much larger) tasks.
>
> References:
> * [R1] Vardan Papyan, 2019, Measurements of Three-Level Hierarchical Structure in the Outliers in the Spectrum of Deepnet Hessians, PMLR.

---

> > ### Comment · Reviewer_S4Ex · 2023-11-23
> >
> > Thanks for your response, I would like to keep my score unchanged.

---

### Official Review · Reviewer_XiKZ · 2023-10-29

**Soundness:** 4 excellent
**Presentation:** 3 good
**Contribution:** 3 good
**Rating:** 8
**Confidence:** 4

**Summary:**

The paper proposes FOSI -  a novel meta-algorithm that improves the performance of any base 1st order optimizer
by efficiently incorporating 2nd-order information.

In each iteration, FOSI implicitly splits the function into two quadratic functions
defined on orthogonal subspaces, then uses a 2nd-order method to minimize
the first, and the base optimizer to minimize the other.

Empirical results shown on many tasks in real-world domain to show the efficacy of POSI.

**Strengths:**

Nice analytics with detailed derivations and explanation.

Large amount of empirical studies shown with experimental results.
Steps of the algorithm are clearly specified.

Enjoyed reading the paper.

**Weaknesses:**

A few failure cases may be discussed.

Although decomposing the problem into two parts may not specifically be novel,
FOSI’s inverse preconditioner seems to be quite a good idea.

Similar work on those lines of decomposing may be mentioned.

**Questions:**

In the statement (step 2m Lemma 1):
"The preconditioner P is symmetric and PD:.

Is it possible to use a different font for the matrix "P", which has no conflict of P (positive) in "PD" ?
or vice-versa.

Hope the code of the paper, to implement FOSI for any application of Optimization,
will be released soon by authors.

---

> ### Author Response · Authors · 2023-11-14
>
> We appreciate the reviewer's thoughtful and positive feedback. Our responses to each comment and query in the review are presented in the same order as they appear in the original review.
>
> **W1:** Thank you for this comment. We do think FOSI provides a novel alternative for hybrid 2nd-order optimizers: rather than estimating the inverse preconditioner (the Hessian), FOSI estimates the preconditioner directly. Beyond this, while we are not aware of any specific optimizer with a similar decomposition followed by application of two different optimizers, we would be glad to include such a paper in our discussion.
>
> **Q1:** We are committed to clear notations and will differentiate more explicitly between the preconditioner’s P and PD for positive definiteness.
>
> **Q2:** Absolutely! The FOSI repository is ready to go, showcasing solid implementations in both Jax and Python. Its API conforms with commonly used packages in these frameworks (Optax and TorchOpt) to facilitate seamless drop-in replacement. Upon the lifting of anonymization requirements, FOSI will be promptly released. You can visit the anonymized repository at https://anonymous.4open.science/r/fosi-497F.

---

### Official Review · Reviewer_dEKh · 2023-10-31

**Soundness:** 3 good
**Presentation:** 3 good
**Contribution:** 2 fair
**Rating:** 5
**Confidence:** 3

**Summary:**

In this paper, the authors present a hybrid method which addresses the challenge of incorporating second-order information in machine learning approaches due to the computational difficulty in high dimensions. This method first splits the space into two orthogonal subspace. Then, the author use first-order method to minimize one of the subspace and use second method to minimize the other one. They also analyze the convergence of FOSI and establish certain conditions under which the proposed method improves the base method. Numerical experiments illustrate the effectiveness of the proposed method.

**Strengths:**

The idea that splitting the raw space into two orthogonal spaces is interesting. The authors adopt the Lanczos to give a possible way to construct these spaces.

**Weaknesses:**

One of my major concern is that the memory consumption and computational complexity are very high especially for large-scale neural networks. This will limit the usage of the proposed method. Besides, it is not clear how to handle the communication cost  and the computation of $V$ in the distributed setting.

The scale of the network architecture used in the numerical experiments is limited. It will be more convincing if the authors can show the effectiveness of the proposed method in larger applications.

**Questions:**

In comparison with second-order methods, it is better to compare the proposed with KLBFGS but not LBFGS because the former method is designed specifically for deep learning tasks. It is suggested to add comparison with Shampoo[1] or NGPlus[2], which has good effectiveness in practice.

it is better to tune the hyper-parameters of ADAM in numerical experiments to make the baseline strong enough.

Why the curve of FOSI-HB decreases in Figure 2?

Can randomized numerical algebra be combined in ESE procedure? This will help reduce the computational cost.

[1] Anil, Rohan, et al. "Scalable second order optimization for deep learning." arXiv preprint arXiv:2002.09018 (2020).
[2] Yang, Minghan, et al. "An efficient fisher matrix approximation method for large-scale neural network optimization." IEEE Transactions on Pattern Analysis and Machine Intelligence 45.5 (2022): 5391-5403.

**Details Of Ethics Concerns:**

No.

---

> ### Author Response · Authors · 2023-11-14
>
> We would like to thank the reviewer for this detailed review. We will address comments and questions in the same order as they appear in the original review.
>
> **W1:** One of the key areas of focus in our future research will be addressing the memory overhead associated with FOSI, as well as parallelization of the algorithm. The first enhancement is to run the ESE procedure on the CPU in parallel to the training process occurring on the GPU. By carefully managing the synchronization between these processes, we aim to reduce the computation bottleneck of computing second-order information and overall training time. This will also reduce FOSI’s memory consumption on the GPU by moving the ESE scratch space memory to the more plentiful CPU memory.
>
> Concerning the execution of the ESE in a distributed setting, our FOSI implementation already supports parallel computation of the dominant $hvp$ operation across multiple GPUs when available (our experiments were performed on a single GPU, however). $hvp$ computation involves gradient computations, allowing for efficient parallelization over the batch axis. We will include this discussion in the final version.
>
> **Q1:** Block diagonal preconditioners such as KBFGS (Goldfarb et al., 2020), Shampoo and its variants [1], and NGPlus [2] have similar limitations as diagonal preconditioners, though to a lesser extent. They may not fully capture the problem's rotation. In contrast, by splitting the problem into two subspaces as done by FOSI, we obtain a full low-rank representation of the Hessian for the first subspace ($\hat{V}$), which captures both the rotation and curvature of the sub-problem. This contributes to accuracy and stability of the optimization, particularly as it is based on extreme eigenvalues and vectors that can be approximated more accurately. We will clarify the relevant discussion in the Related Work section.
>
> In particular, KLBFGS is similar to K-FAC as both algorithms employ a Kronecker factored block-diagonal approximation of the Hessian, and both perform similarly (see Goldfarb et al., 2020). As our experiments show, while K-FAC is indeed better than classic LBFGS, its performance is still substantially worse than SGD, HB, Adam, and their FOSI variants.
>
> Concerning the specific paper by Anil et al. [1], this paper proposes a distributed variant of the Shampoo optimizer (Gupta et al., 2018) that also uses parallel CPU computation. As noted above, while we intend to explore such optimizations for FOSI in our future research, we consider it beyond the scope of the current work.
>
> Finally, we wish to stress that FOSI provides a new alternative formulation for hybrid 2nd-order optimizers: while prior work estimates the inverse preconditioner (the Hessian), FOSI estimates the preconditioner directly. Inverse preconditioner approaches such as K-FAC, K-BFGS and Shampoo have been developed and improved over years of research, and there are now many proposed second-order optimizers based on their principles with various targeted optimizations. We therefore opted to compare our approach with the two most representative ones so that we can focus on the difference between our direct preconditioner approach and inverse preconditioner approaches, rather than specific optimization.
>
> **Q2:** The primary focus of our paper is to demonstrate FOSI's improvement over the base optimizer, without additional tuning, and even when the base optimizer is not perfectly tuned. This aligns with our drop-in replacement concept for FOSI. Rather than tuning Adam, we show in Section 4.3 and Appendix B 2.2, figure 9, that FOSI improves performance across a wide range of learning rate and momentum. Also, given Adam's inherent adaptability, it tends to perform well across a relatively broad spectrum of hyperparameters.
>
> **Q3:** This drop is due to overfitting. Note HB similarly overfits: while we cut the x axis after 7000 seconds for clarity of presentation, we can observe HB starting to overfit after around 5200 seconds. In practice, whenever the base optimizer overfits, FOSI’s acceleration of convergence also leads to overfitting, simply earlier. We describe this phenomenon in the initial paragraph on page 8. Throughout our experiments, we have not observed FOSI to overfit more than the base optimizer; FOSI always reaches the same or superior validation accuracy as the base optimizer.
>
> **Q4:** Thank you for this suggestion. We are actively exploring multiple avenues to enhance the efficiency of ESE computation. It's noteworthy that, in line with the reviewer's suggestion of randomized numerical algebra, our current approach already incorporates elements of this concept. Specifically, we utilize a set of samples to approximate the Hessian in the $hvp$ computation. However, we acknowledge there is considerable potential for further optimization in this direction, including the exploration of techniques such as warm starting the Lanczos algorithm and other more intricate methods.

---

### Official Review · Reviewer_UYzc · 2023-11-01

**Soundness:** 3 good
**Presentation:** 2 fair
**Contribution:** 3 good
**Rating:** 6
**Confidence:** 4

**Summary:**

In this paper, the authors presented a novel optimization method called FOSI, which is a hybrid optimizer algorithm that combines the first-order optimizer with Newton's method. They presented some theoretical analysis as well as some empirical results from the numerical experiments.

**Strengths:**

This submission is well-organized with clear language and structures. The authors gave detailed description and some theoretical analysis for the proposed algorithms. They also conduct a lot of numerical experiments on deep learning problems and these empirical results are pretty good compared with some state-of-art optimization methods. The idea is pretty interesting and enlightens some promising future direction for the optimization community.

**Weaknesses:**

There are some disadvantages regarding this submission.

The authors only gave the theoretical results for the stochastic optimization problem. What's the convergence rate for the general convex optimization problem? What is the convergence rate for the strongly convex setting? If the authors could add and present these theoretical analysis. This could significantly improve the quality of this submission.

It's better to put the detailed algorithm from the appendix to the main part of the paper.

It's better to put the section 5 related work part in the section 1 introduction part.

**Questions:**

Please check the weakness section.

---

> ### Author Response · Authors · 2023-11-14
>
> We appreciate the review and will address comments and questions in the order they were presented.
>
> **W1:** The convergence analysis in the convex setting, encompassing both general and strongly convex scenarios, relies on the choice of the base optimizer. In this context, the convergence rate of the base optimizer concerning the subproblem $f_2$ becomes the limiting factor, given that Newton's method exhibits a quadratic convergence rate on $f_1$. Consequently, for any given base optimizer, FOSI's convergence rate is the same as the convergence rate of the base optimizer for $f$, but with improved constants due to the smaller condition number of $f_2$.  For example, the convergence analysis of GD yield $f^t - f^* \le \|x^0 - x^* \|^2 / (2\alpha t)$ for $\alpha \le 1/L$. In the convex case, $L=\lambda_1$ (the maximal eigenvalue of the Hessian). Since FOSI-GD reduces the maximal eigenvalue to $\lambda_{k+1}$, its bound is tighter.
>
> We will incorporate that discussion into the text.
>
> **W2:** We agree with the reviewer. The pseudo-code was moved to the appendix due to space limitations. We will endeavor to incorporate it back into the main paper for the camera-ready version.

---

### Meta-Review · Area_Chair_GcTb · 2023-12-03

**Metareview:**

The paper combines first and second-order optimization methods by leveraging complementary subspaces that arise as part of the Lanczos process applied to the Hessian of the objective function. The proposed algorithm, called FOSI, splits the function into two quadratic functions defined on orthogonal subspaces: the eigenspace corresponding to the approximation of the extreme eigenvalues of the Hessian, and its orthogonal complement. These two quadratic approximations form the basis for FOSI to obtain its second- and first-order directions, respectively.

Although the theoretical guarantees are relatively straightforward, offer limited novelty, and/or ignore many subtleties, the paper presents some interesting ideas. The numerical examples also demonstrate the effectiveness of the proposed method.

**Justification For Why Not Higher Score:**

The theoretical guarantees are straightforward or offer highly limited novelty. Many subtleties in the theory are overlooked.

**Justification For Why Not Lower Score:**

The idea of combining first and second order directions are interesting and the numerical examples also demonstrate the effectiveness of method.

---

### Decision · Program_Chairs · 2024-01-16

Accept (poster)